# Chemical-mineralogical features and physical properties of archaeological adobe: The evidence from Tell Zurghul/Nigin (Dhi Qar, Iraq)

Luca Volpi[1]*, Francesco Santoro De Vico[2], Anna Arizzi[3], Nicola Lanzaro[4],
Davide Nadali[4]

1 Departamento de Historia Antigua, Historia Medieval y Paleografía y Diplomática, Universidad Autónoma de Madrid, Madrid, Spain, 2 "Giuseppe Colombo" University Center for Space Studies and Activities (CISAS), University of Padova, Padova, Italy, 3 Departamento de Mineralogía y Petrología, Facultad de Ciencias, Universidad de Granada, Granada, Spain, 4 Dipartimento di Scienze dell'Antichità, Sapienza Università di Roma, Roma, Italy

* luca.volpi@uam.es

## Abstract

This study examines the chemical, mineralogical, hydric, and mechanical properties of archaeological adobe bricks and earthen plasters from the site of Tell Zurghul/Nigin in southern Iraq, which are associated with recently excavated buildings dated to the 5th and 3rd millennium BCE. As integral components of earthen architecture, adobe structures are susceptible to rapid deterioration when subjected to environmental conditions and other degradation factors. Consequently, their preservation is a critical focus within cultural heritage initiatives involving earthen constructions. A comprehensive understanding of the materials used in buildings and their inherent properties is essential for identifying optimal conservation methods for archaeological earthen structures. The research is driven by two primary objectives. Firstly, it seeks to explore construction techniques within a diachronic framework, assessing their temporal evolution while considering the geological context and availability of local raw materials. Secondly, mineralogical and chemical analyses, alongside hydric and mechanical tests, aim to evaluate the characteristics of the adobe bricks, including their resistance and durability against weathering.

## Introduction

Once excavated, archaeological earthen masonry is quickly re-exposed to the decay processes it experienced during its period of use [1]. However, whereas in the past earthen structures were originally surface-plastered and protected from precipitation by the roof, after modern excavation, earthen constructions are directly exposed to natural elements without further protection [2].

In this paper, we investigate adobe architecture, considered as part of a larger set of building techniques collectively referred to as earthen architecture (including cob,

**Data availability statement:** All data are available under CC-BY-SA-4.0 license from Figshare (https://doi.org/10.6084/m9.figshare.30933299). Data for the entire EnEAp project are also deposited into the e-cienciaDatos Online Dataverse (https://edatos.consorciomadrono.es/dataverse/EnEAp).

**Funding:** This research was funded by the European Union's Horizon 2020 Research and Innovation Programme under grant agreement No. 101034324 (PI: Luca Volpi), and by the Ministerio de Ciencia, Innovación y Universidades (MICIU), Agencia Estatal de Investigación (AEI; 10.13039/501100011033; PI: Anna Arizzi), and the European Regional Development Fund (ERDF/FEDER, EU) under the project PID2023-146405OB-100 (2024–2027; PI: Anna Arizzi). The funders had no role in study design, data collection and analysis, decision to publish, or preparation of the manuscript.

**Competing interests:** The authors have declared that no competing interests exist.

rammed earth – tapial or pisé –, and wattle and daub; [3–5]). The selected case-study area is West Asia, a region in which the use of earthen masonry is substantial due to the presence of important river basins (i.e., the Euphrates and the Tigris rivers), which participate in creating an environment rich in mud and clay, and due to the climatic conditions of the area. Its widespread presence and cost-effectiveness have made clay one of the main building materials in the area from prehistoric times to the present day [6].

When dealing with the excavation of adobe architecture, several questions arise as to the appropriateness of adopting integrated solutions for their conservation (see, e.g., [7,8]), and current procedures for protecting archaeological earthen structures are mainly based on four approaches: 1) backfilling [1: 79–85, [9–12]; 2) the construction of temporary or permanent protective shelters [1: 117–120, [13–22]; 3) limited interventions on the structures using newly-made locally produced adobe bricks (e.g., for encapsulation and/or capping), by filling voids and cracks with mortar, and by plastering the adobe bricks with a lime-clay-straw mixture [1: 86–91, [23–30] (for the complete reconstruction of a adobe massive structure, see [31]); 4) interventions on the structures using stabilising materials as a hardener (in particular, ethyl-silicate and aqueous and acrylic polymeric solutions have been utilised; [1: 92–97, [32–37]).

The detailed understanding of building materials and their chemical-physical properties and mechanical resistance is as relevant as the identification of the best preservation technique to be used on archaeological earthen structures [38–45]. The present research intends to investigate the properties of adobe bricks and plasters used for building purposes during the 5th and the 3rd millennium BCE at the archaeological site of Tell Zurghul/Nigin (Nasiriyah) in southern Iraq [46].

Two main objectives are recognised: on the one hand, the interest in investigating construction techniques at both a synchronic and diachronic level in an important site of ancient Mesopotamia, together with their consistency with the geological context of the area and the local raw materials. In parallel, the chemical and mineralogical composition of the adobe bricks was systematically analysed, and a series of hydric and mechanical tests were conducted to characterise the physical properties of the materials, including mechanical strength, water absorption behaviour, and resistance to weathering. These results provide a quantitative basis for assessing the durability of the adobe bricks and for informing targeted conservation and preservation strategies at both the site-specific level and across the wider region.

### Archaeological, geological, and climatic context

**Archaeological setting.** Tell Zurghul/Nigin (31°22'36.06"N; 46°29'36.24"E) is an archaeological site located in south-eastern Iraq (Dhi Qar). The site is approximately 40 km north-east of Nasiriyah, the capital city of the Dhi Qar Governorate, and ca. 7 km south-east of the ancient Sumerian city of Lagash, present-day Tell al-Hiba.

Since 2014, an Italian archaeological expedition of Sapienza University of Rome (MAIN: Missione Archeologica Italiana a Tell Zurghul/Nigin) has been regularly working with yearly seasons of excavation.

The settlement covers a surface of about 70 hectares, and it has been uninterruptedly occupied from the mid-5th to the end of the 3rd millennium BCE, when the city was destroyed and never extensively reoccupied. The site was part of the ancient State of Lagash, and, thanks to several cuneiform inscriptions, it has been identified with the Sumerian city of Nigin. The site landscape is characterised by two extensive mounds, called Mound A and Mound B (Fig 1). The first one corresponds to the main occupational phases of the ancient city of Nigin already known by cuneiform sources (3rd millennium BCE; [47–49]). Mound B is located some meters south-west of Mound A, and it is mainly occupied in the prehistoric phases of the site (mid-5th millennium BCE; [50,51]).

**Geology of the area and climatic conditions.** The Nasiriyah region, situated within the Lower Mesopotamian Plain (LMP) of southern Iraq's Dhi Qar Governorate, represents a critical transition zone between the fluvio-deltaic systems of the Tigris-Euphrates rivers and the Persian Gulf [52]. Geomorphologically, the area is characterised by Quaternary alluvial sediments overlying an unstable shelf zone created by the collision of the Arabian plate with the Zagros Fold-and-Thrust Belt [53,54] (Fig 2).

This tectonic setting has resulted in a subsiding basin with a northwest-southeast trending fault system that fundamentally influences hydrological patterns and sedimentation processes. The Nasiriyah area exhibits a complex stratigraphy that reflects multiple depositional environments, including fluvial, lacustrine, and episodic marine influences. Sedimentological evidence, including the Hammar Formation, indicates that marine transgression reached this region during the mid-Holocene [55]. The contemporary landscape manifests a low-gradient topography (approximately 3-5m above MSL) with distinctive anastomosing river patterns as the Euphrates approaches the Hammar Lake zone. This geomorphological configuration has played a crucial role in shaping human settlement patterns since the early 6th millennium BCE, with

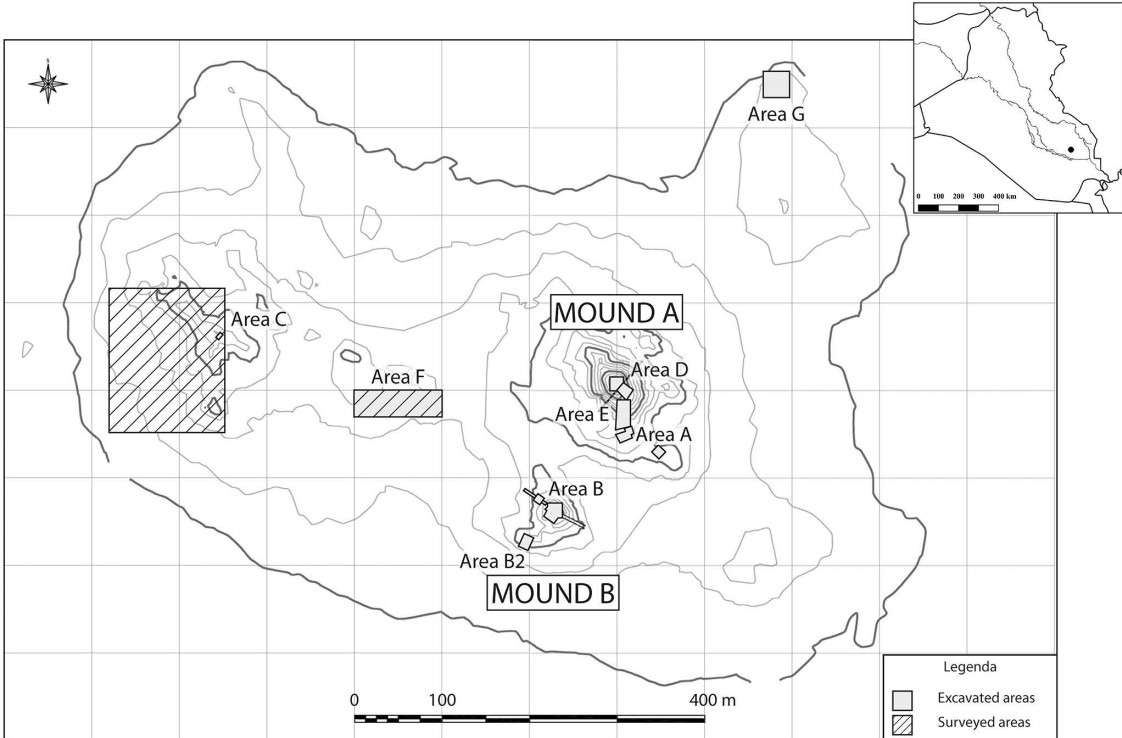

**Fig 1. Topographic map of the Tell Zurghul/Nigin site with the indication of surveyed and excavated areas.** Mound A has been investigated in several operations (Area A, D, and E); Mound B has been investigated in one operation named Area **B.** Reprinted from [46] under a CC BY license.

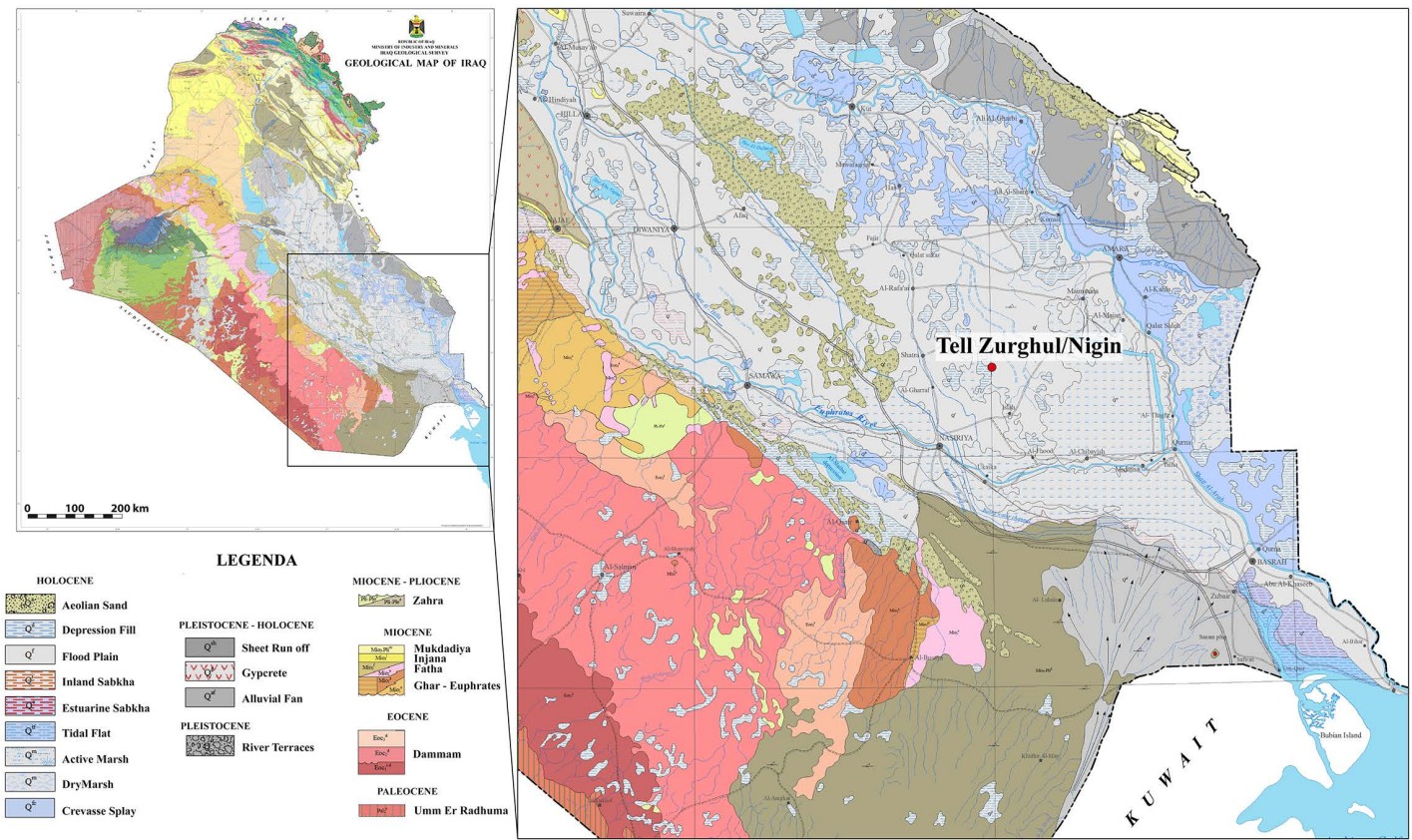

**Fig 2. Geological map of Iraq.** Reprinted from [54] under a CC BY license.

the Nasiriyah area functioning as an interstitial ecological zone where fluvial deposition, climate fluctuations, and human adaptations have interacted throughout the Holocene [56].

Tell Zurghul (Nasiriyah, Iraq) is the selected site spot, characterised by a semi-arid (BWh) climate as classified by Köppen-Geiger (https://koeppen-geiger.vu-wien.ac.at/; [57]).

Southern Iraq features a hot desert climate with extremely hot summers, where temperatures regularly exceed 40°C, particularly in July and August. The region receives minimal annual rainfall, less than 100 mm, mostly occurring between November and April. The rainfall regime is torrential (Fig 3).

## Materials and methods

### Provenience of archaeological samples

Thirty archaeological samples (29 adobe bricks and one earthen plaster sample) were collected during the 2021–2022 excavation seasons at Tell Zurghul/Nigin by MAIN, and exported with the permission of the SBAH (State Board of Antiquities and Heritage) in Baghdad (Table 1).

The selection criteria are based on the desire to investigate construction techniques at a synchronic level and their evolution over time. Therefore, three to four samples for each architectural phase in Mound B and three samples from Mound A have been selected. Only a single sample of earthen plaster has been selected, as it is the sole well-preserved specimen identified to date at Tell Zurghul/Nigin, and it was included primarily to compare its composition and use of local clay sources with the sampled adobe bricks.

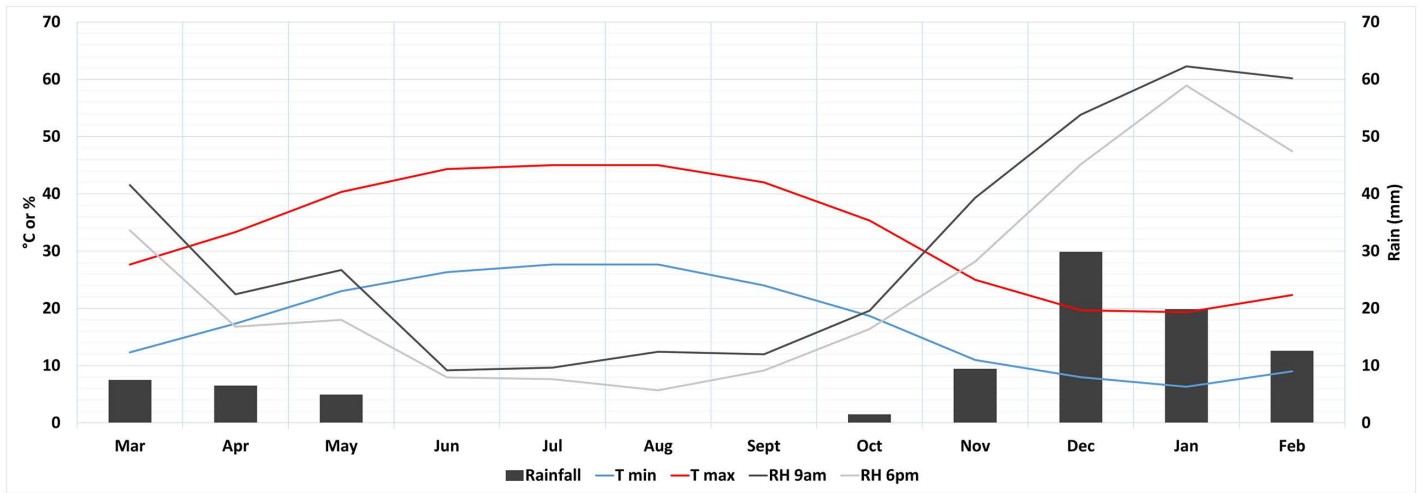

**Fig 3. Tell Zurghul/Nigin real climatic conditions, mean values from years 2021-2011-2001.** Data from the WorldClim database at https://worldclim.org/; S1 Fig for further information).

Twenty-seven samples come from Mound B, and they are dated to the mid-5th millennium BCE. Archaeologically, associated pottery materials are assigned to the Late Ubaid (Ubaid 4) phase. The operations focused on two different portions of Mound B: an extensive area (ca. 20 x 16 m) has been investigated on the top of the mound since 2015 and aimed at the comprehension of the overall surface architectural layouts and their stratigraphic sequence; the second area is a step trench (ca. 30 x 3 m) excavated in 2021 in the northern slope of the mound to investigate the earliest occupation of the mound [50,51]. Nine samples from the operation on top of the mound come from wall structures belonging to different stratigraphic layer and architectural phases (henceforth, AP) 2b, AP 2c, and AP 3a (numerical order is from most recent to earliest; Figs 4A and 5A-B). Twelve samples come from the step trench, belonging to AP 4, AP 5, and AP 6. At the base of the step trench, an archaeological layer dated to the mid-4th millennium BCE (Step V) has been found. Four samples originate from walls associated with this level (Figs 4B and 5C). Two samples come from the excavations in Mound B, but they are not associated with any architectural phase.

Two adobe brick samples and one earthen plaster sample come from Mound A, and they are dated to the mid-3rd millennium BCE. Samples come from Area E, a 15 x 20 m trench, located on the south-western slope of Mound A. The operations of the 2019 and 2022 seasons aimed to detect the occupational nature and sequence of the mound, investigating the morphology of the hill in connection with the northernmost Area D (excavated in 2016 and 2019; [47,49]) and the southernmost Area A at the base of the mound (excavated in 2015, 2016, and 2017; [58,59]). They belong to a wall structure (W.458) dated to the ED IIIB/early Akkadian horizon by pottery (Fig 4C and 5D–E).

## Archaeological and autoptic group distinction

Samples are divided into groups based on their archaeological provenience and their visual distinction. This subdivision was done before any archaeometric analysis to group the samples according to visual features, and based on the following parameters: compactness, texture, and colour, assessed through unaided visual inspection. Six groups have been identified (Fig 6).

Group 1 is characterised by a compact matrix with few visible voids and a buff-light beige colour. Group 2 is distinguished from the former by a darker brown colour. Samples in group 3 have a granular appearance with clay arranged in small lumps, and with a dark beige-greyish colour. Group 4 is very compact, with a more clayey appearance and a light

**Table 1. ID samples, provenience, archaeological phasing, and dating.**

| ID | Wall N. | Arch. Ph. | Arch. Period | Type |
|---|---|---|---|---|
| 43 | W.389 | Mound B, Step V | Uruk | Adobe brick |
| 44 | W.292 | \ | Late Ubaid | Adobe brick |
| 45 | W.386 | Mound B, AP 5 | Late Ubaid | Adobe brick |
| 46 | W.389 | Mound B, Step V | Uruk | Adobe brick |
| 47 | W.381 | Mound B, AP 4 | Late Ubaid | Adobe brick |
| 48 | SU 861 | Mound B, AP 6 | Late Ubaid | Adobe brick |
| 49 | W.389 | Mound B, Step V | Uruk | Adobe brick |
| 50 | SU 861 | Mound B, AP 6 | Late Ubaid | Adobe brick |
| 51 | W.382 | Mound B, AP 4 | Late Ubaid | Adobe brick |
| 52 | W.382 | Mound B, AP 4 | Late Ubaid | Adobe brick |
| 53 | W.389 | Mound B, Step V | Uruk | Adobe brick |
| 54 | W.386 | Mound B, AP 5 | Late Ubaid | Adobe brick |
| 55 | W.381 | Mound B, AP 4 | Late Ubaid | Adobe brick |
| 56 | W.361 | Mound B, AP 2c | Late Ubaid | Adobe brick |
| 57 | W.381 | Mound B, AP 4 | Late Ubaid | Adobe brick |
| 58 | SU 861 | Mound B, AP 6 | Late Ubaid | Adobe brick |
| 59 | W.386 | Mound B, AP 5 | Late Ubaid | Adobe brick |
| 60 | W.382 | Mound B, AP 4 | Late Ubaid | Adobe brick |
| 61 | W.362 | Mound B, AP 2b | Late Ubaid | Adobe brick |
| 62 | W.264 | Mound B, AP 2b | Late Ubaid | Adobe brick |
| 63 | W.362 | Mound B, AP 2b | Late Ubaid | Adobe brick |
| 64 | W.701 | Mound B, AP 3a | Late Ubaid | Adobe brick |
| 65 | \ | \ | Late Ubaid | Adobe brick |
| 66 | W.701 | Mound B, AP 3a | Late Ubaid | Adobe brick |
| 67 | W.361 | Mound B, AP 2c | Late Ubaid | Adobe brick |
| 68 | W.361 | Mound B, AP 2c | Late Ubaid | Adobe brick |
| 69 | W.701 | Mound B, AP 3a | Late Ubaid | Adobe brick |
| 79 | W.458 | Mound A | Mid-3rd Mill. BCE | Adobe brick |
| 80 | W.458 | Mound A | Mid-3rd Mill. BCE | Adobe brick |
| SG.22.E.302/9 | W.458 | Mound A | Mid-3rd Mill. BCE | Plaster |

beige colour. Group 5 shows a higher appearance of sand and organic inclusions (shell). Group 6 is composed of all 3rd millennium BCE samples (both adobe bricks and earthen plaster) with an orange colour (Table 2).

## Characterisation and testing methods

**Chemical, mineralogical, and petrographic study.** 5g of each sample was powdered in an agate mill to perform chemical and mineralogical analysis (30 samples).

Chemical analysis was carried out by X-ray fluorescence (XRF) with an XRF energy disperser spectrometer PANalytical Zetium, which integrates a tube with a rhodium anode, an X-ray generator of 4 kW, a decoupled goniometer θ/2θ, and proportional detectors of scintillation and high-level flow. The analysis was performed on all samples and was used to determine and quantify major and trace elements in the samples' composition [60: 331–399].

Mineralogical analysis was performed by Powder X-ray diffraction (PXRD) using a X'Pert Pro Diffractometer equipped with a continuous sample rotation system using the disordered crystalline powder method. The experimental conditions

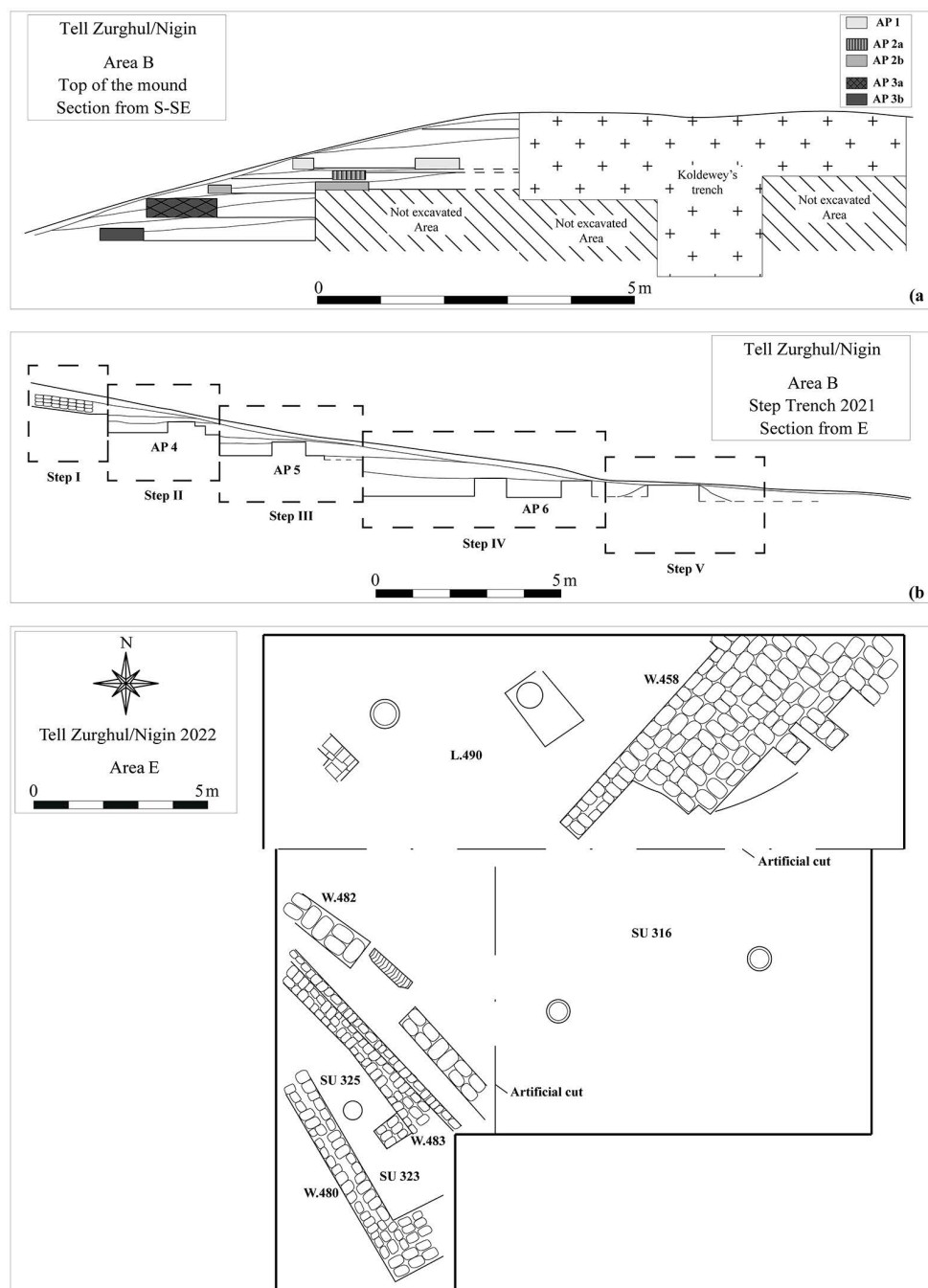

**Fig 4. Graphic documentation of the excavation areas.** A) Area B: reconstructed section of the excavation at the top of the mound (APs 1–3); B) Area B: reconstructed section of Step Trench excavation in 2021 (APs 4–6); C) Area E: plan of the 2022 excavation season. Reprinted from [46] under a CC BY license.

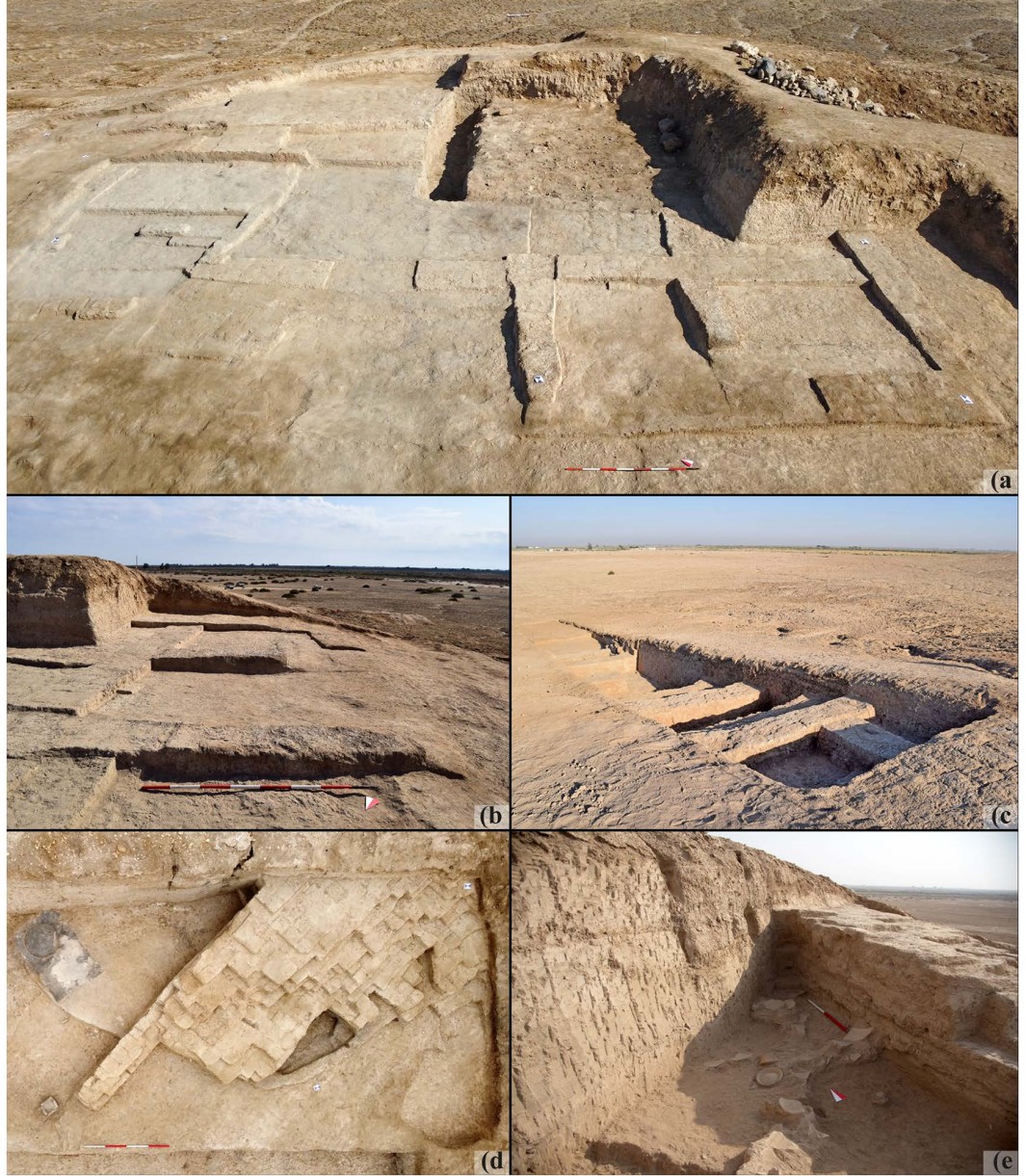

**Fig 5. Photographic documentation of the excavation areas.** A) Area B: aerial view of the excavation at the top of the mound (APs 1–3); B) Area B: detail of adobe wall remains at the top of the mound (APs 1–3); C) Area B: picture of the Step Trench excavation in 2021 (APs 4–6), from south-west; D) Area E: aerial view of W.458; E) detail of the inner face of W.458 entering the excavation section.

were as follows: voltage of 45 kV; current of 40 mA; CuKα radiation (λ = 1.5405 Å); explored area between 4° and 70° 2θ; goniometer speed of 0.1° 2θ/s. The results were analysed with the X'Pert Highscore Plus 4.9 software (PANalytical), equipped with the Joint Committee for Powder Diffraction Standards (JCPDS) database. The analysis was performed on all 30 samples and was used to determine their crystalline structures [60: 430–438].

Polished thin-sections of 8 (out of 30) adobe brick samples and one earthen plaster sample were analysed under a polarized optical microscopy Carl Zeiss "Jenapol-U" equipped with a Nikon D7000 reflex camera. The selection is

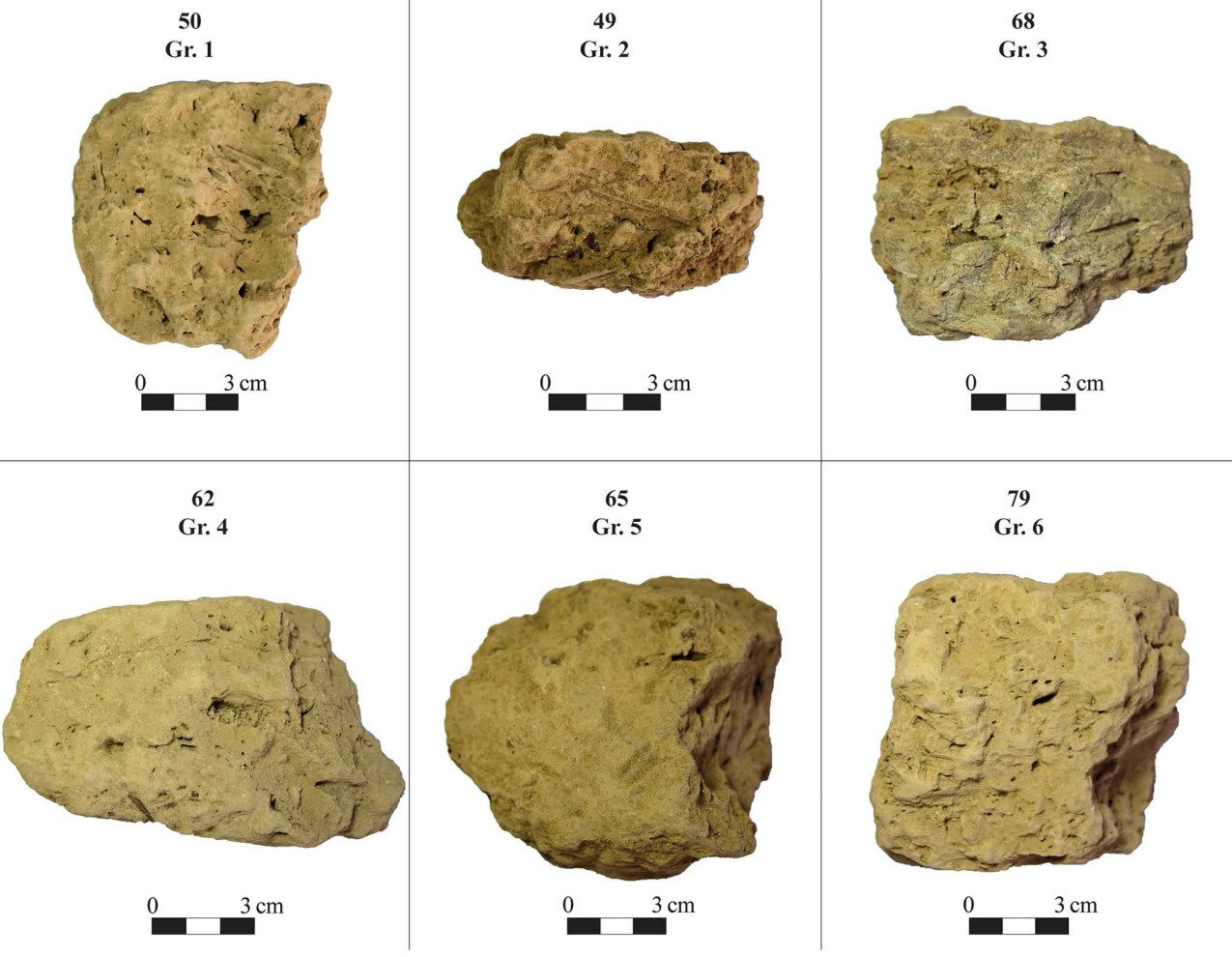

**Fig 6. Adobe brick samples (photos) for each autoptic group (1–6).**

**Table 2. Distribution of samples across autoptic groups.**

|  | Group 1 | Group 2 | Group 3 | Group 4 | Group 5 | Group 6 |
|---|---|---|---|---|---|---|
| ID | 43, 44, 45, 46, 47, 48, 50, 54, 55, 57, 58, 59, 60, 66, 69 | 49, 53, 56, 67 | 51, 52, 68 | 61, 62, 63 | 64, 65 | 79, 80 302/9 |

based on a multitude of factors, which include, among others: the size of the original sample to obtain a thin section that would allow sufficient material to be preserved for further tests; the attempt to cover, as broadly as possible, the spectrum of macroscopic differences recognised in the differentiation into autoptic groups; the inclusion of two samples dating to the 3rd millennium BCE (as opposed to the others dating to the 5th millennium BCE). The samples were stabilised with epoxy resin before being cut to produce polished thin sections. The petrographic characterisation was used to determine texture, granulometry, porosity, nature of clay and mineral components, and presence/absence of organic materials [60: 21–45].

**Colour.** A spectrophotometer CM-700d Konica Minolta, with a 3 mm diameter (SAV), and a CM-S100W DATA Software COLOR SpectraMagic NX was used for the colorimetry test. Colour analysis was performed on the original samples (30 samples), without further preparation.

Colour properties of the samples were registered following the EN 15886:2010 standard. The obtained data is expressed using the chromatic parameters of the CIELAB system: L* (luminosity), a* (green to red spectrum), b* (blue to yellow spectrum), h* (tone/shade), and C* (chroma).

**Compactness and strength.** Compactness and strength tests (Scotch Tape Test and Surface Hardness Test) were performed on the original samples (30 samples), without further preparations.

The Scotch Tape Test (or Peeling Test) was performed to assess the surface cohesion of the samples, and to provide an indication of the degree of surface deterioration of the adobe bricks and plaster.

The detached material from each sample was analysed through visual inspection (photos) and weight measurement, following [61]. The weighting was conducted based on ASTM D3359 regulation, with a ±0,0001 g precision scale. 2.6 x 3.1 paper labels (ca. 8 cm$^2$) with double-sided tape were prepared and weighted (unladen weight). The paper labels were attached to a flat sample face (both surface and core) with homogeneous pressure (up to six pressures, adhesion time: 0.30 min). The labels were stripped off (tear-off angle of ca. 90°) and re-weighed to quantify the amount of material remaining adhered on the tape. Two to three measurements were taken on each sample, depending on its nature and preservation.

The Surface Hardness test was performed with a portable hardness tester with a striker PCE-2500N (PCE Ibérica). Measurements were taken directly from the surfaces of the adobe bricks and plaster, when available (26 samples; among the other four, two are too small in size to apply the measurement, and two have values below the instrument's minimum threshold). Ten measurements were taken for each sample in order to have representative values and an average.

**Hydric and rain resistance tests.** Surfaces and horizontal fresh cut sherds of samples were prepared for static contact angle and imbibition test (30 samples).

The static contact angle (CA) of samples was measured by the sessile drop method (4 μL of MilliQ® water) on a Data Physics OCA 15EC device equipped with a Peltier stage (25 °C) and SCA20 (Version 5.0.41) software (see [62]). Two measurements for each sample were performed, one on the original surface (R), and one on a fresh-cut fracture of the core (F).

The imbibition time test of both real and fresh-cut surfaces has been measured through video analysis.

Both static contact angle (CA) and imbibition time test have also been performed on 5 cubes with a side of ca. 2 cm used for the rain resistance test (see below). Tests have been performed on two surfaces for each cube.

Rain resistance tests are carried out to evaluate the impact of environmental conditions on various materials (i.e., adobe bricks) in an accelerated time (see, e.g., [63]). The test enables the replication of specific climatic conditions within a controlled laboratory environment, allowing the decay process to be parameterised over a defined time period. The analysis was performed to simulate the environmental conditions (in particular rainfall) to which adobe structures are exposed in Southern Iraq.

The temperature ramp was applied during each wetting period, with the test conducted in a precision oven "Digitronic-TFT", SELECTA®, maintaining controlled temperature ranges (from room temperature to an average of the maximum temperatures recorded in the area in years 2001-2011-2021 as depicted in Fig 3, to expose the samples to maximum thermal stress for each cycle).

One year of rainfall variations was simulated over four days, with each season represented by a one-day test, thus simulating a one-year stress. Before each daily cycle, the corresponding seasonal rainfall was applied using a pipette. Water application methods include immersion, drop-by-drop with a pipette, or using a spray, each representing different rainfall intensities. Considering the rainfall characteristics in Southern Iraq, derived from differential precipitation regimes and intensity-duration-frequency (IDF) curves [64], Nasiriyah experiences torrential rain. In order to simulate torrential

rain, pouring all the water at once is preferable. We used a 3 mL pipette, with each drop weighing approximately 50 mg (0.05 mL), to apply the entire quantity of water in a short period of time in order to simulate torrential rain (S1 Table). Relative humidity was not set as a fixed value, as it varied as a function of the temperature tested in each period/season.

Pre-wetting and wetting conditions were monitored using weight measurements, photogrammetry, and caliper measurements. Data were registered at the start of each daily cycle.

Climatic data were sourced from the WorldClim database (https://worldclim.org/) and plotted using QGIS to collect climatic data on specific spots (30-second image resolution).

Rainfall (mm; Fig 3) per month was recorded for three reference years (2001, 2011, and 2021). Average monthly seasonal precipitation for the three reference years (aggregated for three months per season) was calculated for the simulation.

Cubes of ca. 2 cm per side were prepared for the rain resistance test (5 samples, nos. 50, 52, 55, 62, and 79). To calculate the amount of water to be applied, the rainfall in millimetres was converted into volume. Each millimetre of rainfall equals 1 litre per square meter (1 L/m²), or 1,000 millilitres per 10,000 square centimetres (mL/10,000 cm²). Since 1 square meter (m²) = 10,000 square centimetres (cm²), the volume of water per square centimetre is 0.1 mL per square centimetre (mL/cm²). The final formula to calculate the water volume V (in mL) to apply to a surface A (cm²) is:

$$V(mL) = \frac{h(mm) \times A\left(cm^2\right)}{10}$$

where h is the rainfall depth in millimetres (mm), and A is the area in square centimetres (cm²).

## Results

### Chemical, mineralogical, and petrographic study

**X-ray fluorescence (XRF).** XRF data indicate that the chemical composition of all analysed samples is relatively uniform (Table 3).

Variations are limited to a few percentage points and appear to result from random factors rather than deliberate ancient technical or cultural choices. Based on abundance criteria, the raw materials used seem to appear consistently the same and locally sourced.

Silicon (expressed as $SiO_2$ in Table 3) is among the most abundant elements, with variances between 38 and 44%. It is mainly found in quartz, feldspars, and other silicates, minerals present in significant quantities in all samples, as discussed in section 4.2.2.

Calcium oxide (CaO) mostly derives from calcite and dolomite minerals, both identified in the mineralogical analysis, while magnesium oxide (MgO) is predominantly found in minerals such as dolomite and serpentine. Sodium oxide ($Na_2O$) is abundant in minerals of the feldspar group, particularly plagioclases.

The Principal Component Analysis (PCA) was carried out on the chemical data through Python 3.8+ (The code used in this study is available at Figshare: https://doi.org/10.6084/m9.figshare.31293067). PCA shows that most samples (except samples nos. 56 and 67) fall within the 95.5% variation, suggesting general uniformity in chemical composition (Fig 7). Some sub-groupings can be identified: the largest group of samples shows a relationship between $Na_2O$ and MgO variables; samples nos. 61, 62, 63, 64, and 69 (in the bottom left quadrant of the plot) show higher CaO content; some samples (in the bottom right quadrant of the graph), including samples nos. 56 and 67, show higher $Al_2O_3$ and $K_2O$ content; the earthen plaster sample no. 302/9 is isolated from the other samples by its higher $Fe_2O_3$ content.

**Powder X-ray diffraction (PXRD).** The mineralogical composition of the samples suggests the same consistency in raw material procurement and use, as already indicated by XRF analysis (Table 4).

Quartz ($SiO_2$) and calcite ($CaCO_3$) are the predominant mineralogical phases in all samples (Fig 8).

 

**Table 3. Chemical composition of the 30 studied samples by means of X-Ray Fluorescence (XRF). The oxide content is given in (%).**

| Sample | SiO2 | Al2O3 | Fe2O3 | MnO | MgO | CaO | Na2O | K2O | TiO2 | P2O5 | Zr | LOI |
|---|---|---|---|---|---|---|---|---|---|---|---|---|
| 43 | 41.72 | 9.42 | 5.16 | 0.10 | 5.98 | 14.84 | 2.74 | 1.59 | 0.60 | 0.28 | 107.8 | 17.56 |
| 44 | 39.04 | 10.09 | 5.03 | 0.09 | 4.35 | 16.64 | 2.07 | 1.96 | 0.58 | 0.14 | 99.0 | 19.99 |
| 45 | 38.40 | 9.16 | 5.21 | 0.09 | 6.01 | 14.23 | 4.16 | 1.49 | 0.59 | 0.15 | 103.5 | 20.50 |
| 46 | 41.72 | 9.49 | 5.20 | 0.09 | 6.05 | 14.53 | 2.55 | 1.63 | 0.62 | 0.24 | 106.1 | 17.87 |
| 47 | 42.01 | 9.72 | 5.35 | 0.10 | 5.87 | 13.98 | 2.75 | 1.61 | 0.60 | 0.15 | 101.5 | 17.85 |
| 48 | 39.80 | 9.60 | 5.50 | 0.10 | 6.45 | 14.61 | 2.88 | 1.57 | 0.63 | 0.15 | 102.7 | 18.68 |
| 49 | 40.56 | 9.33 | 5.22 | 0.09 | 6.41 | 14.54 | 2.53 | 1.66 | 0.60 | 0.20 | 105.8 | 18.85 |
| 50 | 40.40 | 9.45 | 5.54 | 0.10 | 5.88 | 14.76 | 2.63 | 1.58 | 0.62 | 0.14 | 109.1 | 18.88 |
| 51 | 40.02 | 11.00 | 5.83 | 0.10 | 5.33 | 14.48 | 1.57 | 2.17 | 0.61 | 0.15 | 98.4 | 18.72 |
| 52 | 39.12 | 10.55 | 5.53 | 0.10 | 5.14 | 15.57 | 1.80 | 2.02 | 0.59 | 0.14 | 97.4 | 19.03 |
| 53 | 40.65 | 9.28 | 5.14 | 0.09 | 6.50 | 14.11 | 2.67 | 1.65 | 0.60 | 0.19 | 105.4 | 19.10 |
| 54 | 41.77 | 9.72 | 5.46 | 0.10 | 6.22 | 15.10 | 2.10 | 1.56 | 0.63 | 0.17 | 114.2 | 16.89 |
| 55 | 42.06 | 9.65 | 5.32 | 0.09 | 5.81 | 14.27 | 2.78 | 1.58 | 0.60 | 0.14 | 98.0 | 17.68 |
| 56 | 42.17 | 12.89 | 6.77 | 0.11 | 5.67 | 10.58 | 2.25 | 2.59 | 0.66 | 0.14 | 108.2 | 16.16 |
| 57 | 41.93 | 9.87 | 5.57 | 0.10 | 6.04 | 14.22 | 2.35 | 1.64 | 0.63 | 0.15 | 102.3 | 17.47 |
| 58 | 39.45 | 9.47 | 5.40 | 0.10 | 6.33 | 14.85 | 2.97 | 1.57 | 0.60 | 0.15 | 102.0 | 19.09 |
| 59 | 40.98 | 9.74 | 5.59 | 0.10 | 6.30 | 15.20 | 2.25 | 1.55 | 0.65 | 0.15 | 118.8 | 17.36 |
| 60 | 38.62 | 10.32 | 5.41 | 0.10 | 5.16 | 16.87 | 1.75 | 1.95 | 0.56 | 0.14 | 93.7 | 18.54 |
| 61 | 43.44 | 8.86 | 4.12 | 0.08 | 4.99 | 15.99 | 2.38 | 1.63 | 0.54 | 0.31 | 98.0 | 17.07 |
| 62 | 43.70 | 8.84 | 4.10 | 0.09 | 4.96 | 15.43 | 2.79 | 1.62 | 0.53 | 0.33 | 98.0 | 17.43 |
| 63 | 43.69 | 8.76 | 4.00 | 0.08 | 4.95 | 15.98 | 2.55 | 1.58 | 0.53 | 0.25 | 103.3 | 17.62 |
| 64 | 44.71 | 8.89 | 4.04 | 0.08 | 4.85 | 15.67 | 2.53 | 1.57 | 0.54 | 0.29 | 98.7 | 16.81 |
| 65 | 38.32 | 8.40 | 4.19 | 0.09 | 5.58 | 17.31 | 2.65 | 1.70 | 0.48 | 0.32 | 101.0 | 20.94 |
| 66 | 43.66 | 9.41 | 4.68 | 0.10 | 4.98 | 16.07 | 1.94 | 1.71 | 0.56 | 0.27 | 104.3 | 16.21 |
| 67 | 42.23 | 12.47 | 6.60 | 0.11 | 5.46 | 11.72 | 1.90 | 2.50 | 0.65 | 0.15 | 107.0 | 15.61 |
| 68 | 40.47 | 11.53 | 6.16 | 0.10 | 5.39 | 14.42 | 1.89 | 2.28 | 0.59 | 0.14 | 97.2 | 16.48 |
| 69 | 44.19 | 8.70 | 3.95 | 0.08 | 4.85 | 15.99 | 2.51 | 1.55 | 0.52 | 0.30 | 100.5 | 16.77 |
| 79 | 40.55 | 9.99 | 5.57 | 0.10 | 6.07 | 15.71 | 1.73 | 1.87 | 0.61 | 0.16 | 97.3 | 17.14 |
| 80 | 41.25 | 10.14 | 5.71 | 0.10 | 6.10 | 15.38 | 1.57 | 1.88 | 0.63 | 0.15 | 98.9 | 16.67 |
| SG22E302/9 | 40.05 | 9.93 | 5.55 | 0.10 | 5.98 | 14.55 | 2.17 | 1.98 | 0.61 | 0.21 | 100.6 | 18.85 |

As the analysis performed was qualitative rather than quantitative, it is currently not possible to discern whether differences in quartz and calcite peaks correspond to actual differences in raw materials use or are due to random factors (such as the selection of one portion of the sample to be powdered instead of another, due to sample heterogeneity). The abundance of Ca, previously observed by XRF, is justified by the presence of dolomite ($CaMg(CO_3)_2$) in almost all samples.

The presence of halite (NaCl) can be explained by soil characteristics. In southern Mesopotamia, the soil is subject to continuous salinisation processes [65]. This phenomenon is documented in cuneiform texts for the 3rd and 2nd millennium BCE [66–70]. Therefore, the presence of halite may be attributed to post-depositional factors rather than its deliberate use in adobe brick production.

Plagioclase feldspars (((Na, Ca) (Si, Al)$_4$O$_8$) are present in almost every sample, predominantly in their calcium-rich mineralogical phase as anorthite ($CaAl_2Si_2O_8$).

Gypsum ($CaSO_4 \cdot 2H_2O$) is another mineral consistently found in almost all samples. It occurs naturally in southern Mesopotamia; its presence can therefore be explained by its local occurrence, while it remains to be determined whether it is present only residually or was deliberately added to the adobe production recipe.

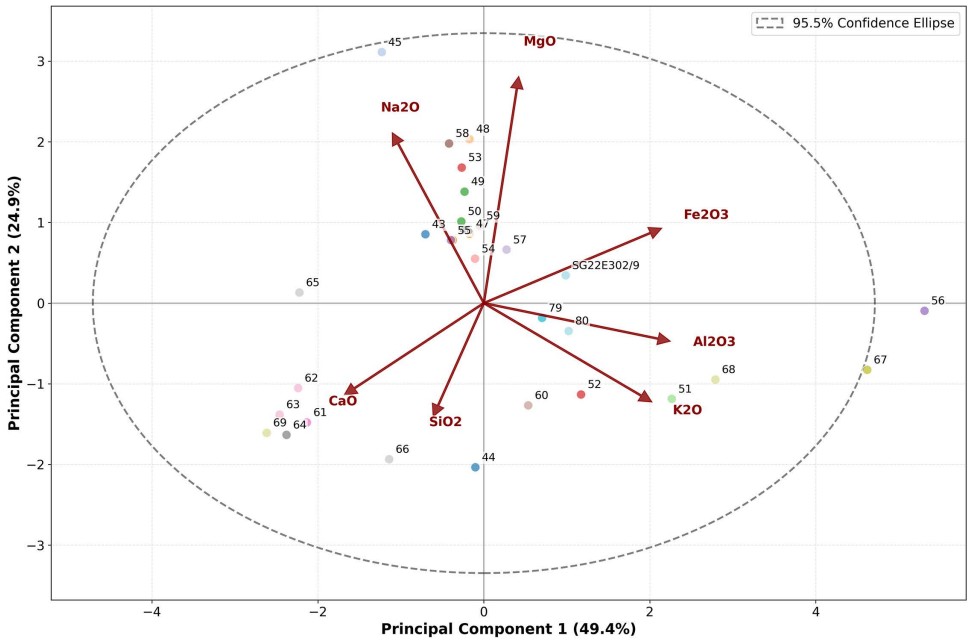

**Fig 7. Principal Component Analysis (PCA) carried out on the XRF data.**

Phyllosilicate minerals of the mica group (in particular, muscovite), and of the chlorite group (clinochlore) are also commonly attested in all analysed samples.

Some mineralogical phases are rarer and not present in all analysed samples (amphiboles, serpentine, sillimanite, palygorskite, orthopyroxene, clinopyroxene, and rutile). Notably, minerals from the amphiboles group (hornblende, as seen in the XRD plots) are attested in 9 of 30 samples; interestingly, they appear in samples nos. 61, 62, 64, and 69, most of which form a subgroup in the chemical PCA.

**Polarized optical microscopy (POM).** Petrographic analysis of thin sections from nine samples (8 adobe bricks and one plaster) revealed distinct textural and mineralogical characteristics (Table 5; Fig 9; S2 Fig).

The primary mineral assemblage consists of quartz, calcite, feldspars, and dolomite as inclusions, while mica, chlorite, and serpentine constitute the binding phases. The samples exhibit varying porosity patterns, with some showing vertically oriented pores in the upper portions and more rounded, interconnected pores in the core. Several specimens display a clear structural differentiation between surface and core regions, with denser matrices observed near the surfaces.

Sample no. 54 exhibits a distinct division between a left zone containing abundant inert materials in the mud crack mixture and a right zone with a high percentage of clay binder.

Sample no. 62 stands out for its high aggregate content and minimal matrix, featuring predominantly crystalline inclusions with angular edges. Halite was identified in multiple samples, particularly on surfaces, possibly indicating weathering processes, as evidenced by its absence in the better-preserved sample no. 62. The matrix-to-aggregate ratio varies significantly across samples, with specimens like nos. 68 and 79 showing clay-rich, matrix-dominated compositions with sparse crystalline inclusions. Sample no. 68 also displays multiple fractures with two preferential fracture orientations. A notable feature in the earthen plaster sample no. 302/9 is the presence of extensive interconnected elongated pores and fractures throughout the section, resembling root patterns. The porosity characteristics vary from compressed bi-dimensional pores with diagonal cross-orientation (sample no. 52) to more homogeneous circular voids (sample no. 54), reflecting different manufacturing techniques or post-depositional processes. Matrix composition and density show

**Table 4. Mineralogical composition of the 30 studied samples obtained by qualitative Powder X-ray Diffraction (PXRD).**

| ID | Quartz | Calcite | Halite | Feldspars (s.l.) Plagioclases | Gypsum | Mica | Chlorite | Dolomite | Amphiboles | Serpentine | Sillimanite | Palygorskite | Orthopyroxene | Clinopyroxene | Rutile |
|---|---|---|---|---|---|---|---|---|---|---|---|---|---|---|---|
| 43 | X | X | X | X | X | X | X | X | X | X | | | | | |
| 44 | X | X | X | – | X | X | X | X | | | | | | | |
| 45 | X | X | X | X | X | X | X | X | X | X | | | | | |
| 46 | X | X | | X | X | X | X | X | | | | | | X | |
| 47 | X | X | X | X | X | X | X | X | X | X | | | | | |
| 48 | X | X | X | X | X | X | X | X | X | | | | | | |
| 49 | X | X | X | X | X | X | X | X | | | | | | | |
| 50 | X | X | X | X | | X | X | X | | X | | | | | |
| 51 | X | X | X | X | X | X | X | X | | | | X | | | |
| 52 | X | X | X | X | X | X | X | X | | | | | | | |
| 53 | X | X | | X | | X | X | X | | | X | | | | |
| 54 | X | X | X | X | | X | X | X | | X | | X | | | |
| 55 | X | X | X | X | X | X | X | X | | | | | | | |
| 56 | X | X | X | X | X | | X | | | | | X | | | |
| 57 | X | X | X | X | X | | X | X | | | | X | | | |
| 58 | X | X | X | X | X | X | X | X | | | | | | | |
| 59 | X | X | X | X | X | X | X | X | X | | | X | | | |
| 60 | X | X | X | X | X | X | X | X | | | | | | | |
| 61 | X | X | X | | X | X | X | X | X | | | | | X | |
| 62 | X | X | | | X | X | X | X | X | | | X | | | |
| 63 | X | X | X | X | X | X | X | X | | | | X | | | |
| 64 | X | X | X | X | X | X | X | X | X | X | | | | X | X |
| 65 | X | X | X | X | X | X | X | | | X | | | | X | |
| 66 | X | X | X | X | X | X | X | | | | | | | X | |
| 67 | X | X | X | X | X | X | X | | | | | | | | |
| 68 | X | X | X | X | X | | X | | | | | X | | | |
| 69 | X | X | | X | X | X | X | | X | X | X | | | X | |
| 79 | X | X | | X | X | X | X | X | | | | | | | |
| 80 | X | X | X | X | X | X | X | X | | | | | | | |
| 302/9 | X | X | X | X | | X | X | X | | | | | | | |

lateral variations within individual samples, particularly evident in sample no. 54, where the left side contains significantly more grains compared to the matrix-rich right side.

## Colour

Colour measurements are relevant for parametrising visual properties of archaeological materials to develop future conservation strategies.

The obtained results show that most samples share similar chromatic characteristics (S2 Table). Lightness (L* parameter; Fig 10A) is generally ranged between mean values of 50 and 65%, with only two exceptions: sample no. 53 (mean value 47.27%) and sample no. 51 (mean value 65.99%).

The a* (green to red spectrum) and b* (blue to yellow spectrum) values are comprised between 3–7 a* (mean) and 13–20 b* (mean). Values are evenly distributed along a median line (Fig 10B), with samples nos. 61, 62, 63, 64, 68, and 69 standing slightly apart in the lower left portion of the median line (having a* mean values comprised between 3–4,

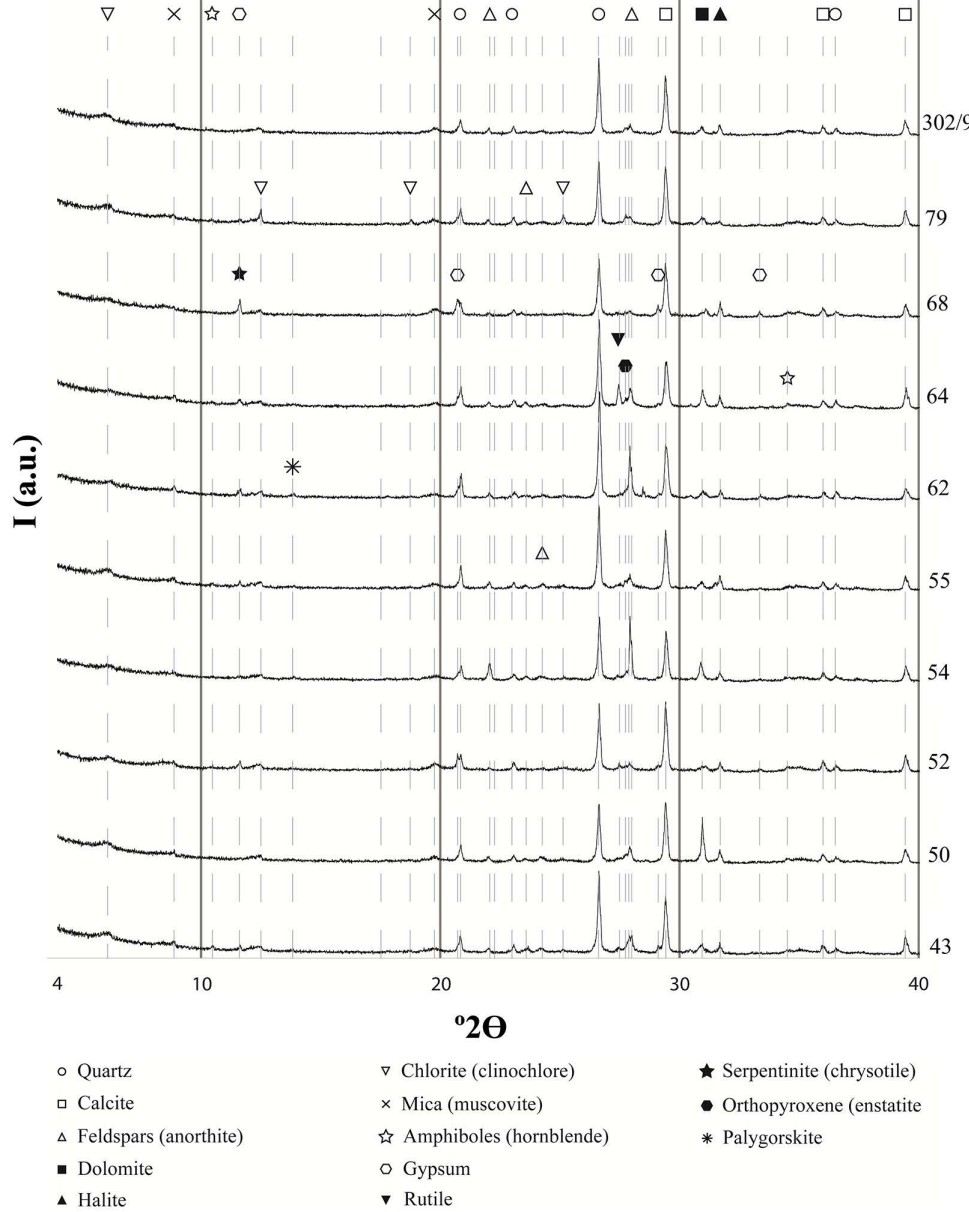

**Fig 8. X-ray Diffraction patterns of 10 samples with indication of the main mineral phases identified.**

and b* mean values comprised between 13–16), in contrast to the other samples, which are distributed between 4.4–6.8 a* (mean), and 14.8–19.1 b* (mean).

## Surface compactness and strength

Results of the Scotch Tape Test, measured in $mg/cm^2$, show varying degrees of material loss after adhesive tape application (Fig 11A).

Samples exhibiting a material loss between 0–2 $mg/cm^2$ demonstrate the best surface cohesiveness, including samples nos. 44, 47, 49, 56, 59, 61, 63, and earthen plaster sample no. 302/9. On the contrary, samples showing more

**Table 5. Petrographic characteristics of samples from archaeological context based on polarized optical microscopy analysis.**

| Sample | Primary Mineral Assemblage | Binding Phases | Matrix-Aggregate Ratio | Porosity Characteristics | Structural Features | Weathering/Special Notes |
|---|---|---|---|---|---|---|
| 43 | Calcite, quartz, feldspars, gypsum, dolomite, amphiboles | Serpentinite, mica, chlorite | Variable: matrix-enriched in the central portion | Upper section: vertical pores. Core: rounded, interconnected pores | Oriented feldspars in core. Dense matrix at the surface. Calcite and feldspar accumulation around pores | Halite is present on the surface. Large clay and mica inclusion |
| 50 | Calcite, feldspars, quartz, dolomite | Chlorite, mica, serpentinite | Matrix-rich | Surface: few horizontally oriented pores. Middle: large horizontal pores. Lower: small, rounded horizontal pores | Clay-enriched zones. Clay aggregates. Interconnected porosity in the central zone | Presence of large clay nodules |
| 52 | Quartz, calcite, feldspars, gypsum, dolomite | Mica, chlorite | High clay content, minimal inert material | Upper section: compressed bi-dimensional pores with diagonal cross-orientation. Central section: dense with large voids. Core: small chaotic pores and large interconnected voids | Bipartite structure: left zone highly compact, right zone with large voids. Large clay aggregates | Structurally weak matrix. Clay aggregates without inclusions connected by fractures |
| 54 | Quartz, calcite, halite, feldspars, dolomite | Mica, chlorite, serpentinite, palygorskite | Variable: left section grain-rich, right section fine matrix-dominant | Relatively homogeneous, circular porosity. Lower section with smaller voids and denser structure | Distinct lateral division: left side with numerous grains, right side matrix-rich. Porosity along the interface between zones | Halite present |
| 62 | Quartz, calcite, gypsum, mica, dolomite, amphiboles, palygorskite | Minimal chlorite | Very high aggregate content, minimal matrix | Large pores in the upper section with east-west orientation. Very limited microporosity | Central fracture. Homogeneous inert material distribution. Angular crystalline inclusions | Absence of halite: potentially less weathered specimen. Angular crystals possibly intentionally crushed |
| 64 | Quartz, calcite, feldspars, gypsum, mica, dolomite, amphiboles, serpentinite, orthopyroxenes | Chlorite | Higher matrix content than sample 62, but still aggregate-rich | Predominantly circular pores. Few but extensive oblique northeast-southwest fractures | Relatively homogeneous structure. Highly crystalline | Localized clay-enriched zones. Potential rutile presence (requires SEM verification) |
| 68 | Quartz, calcite, halite, feldspars, gypsum | Chlorite, palygorskite | Very high matrix content, minimal aggregates | Limited, concentric porosity | Extensive horizontal and vertical fracturing. Laminated structure | Halite present |
| 79 | Quartz, calcite, feldspars, gypsum, mica, dolomite | Chlorite | High matrix content, limited inert material | Oriented voids | Larger plagioclase and quartz inclusions are dispersed in the matrix. Globular masses of feldspar and quartz. Plagioclase twinning | Mineral nodules present |
| 302/9 | Quartz, calcite, halite, feldspars, mica, dolomite | Chlorite | Relatively low aggregate to matrix ratio | Numerous elongated, interconnected pores. Few small pores. Multi-directional orientation | Extensive interconnected fractures throughout the section. Quartz and feldspar nodules | Root-like pattern of interconnected elongated porosity. Halite present |

Note: All microphotographs were taken with north-south orientation.

disaggregated surfaces and, therefore, major loss of adhered materials are nos. 46, and 55 (material loss > 7 mg/cm$^2$). The remaining samples exhibit material loss between 2–6 mg/cm$^2$.

Surface cohesiveness can be correlated with Surface Hardness test data (Fig 11B). In some cases, the preserved sample surfaces were insufficient to perform the test (samples nos. 44, and 47) or not hard enough to withstand the striker percussion (samples nos. 50, and 64, from which no results could be retrieved; S3 Table). Mean values range from 181.8

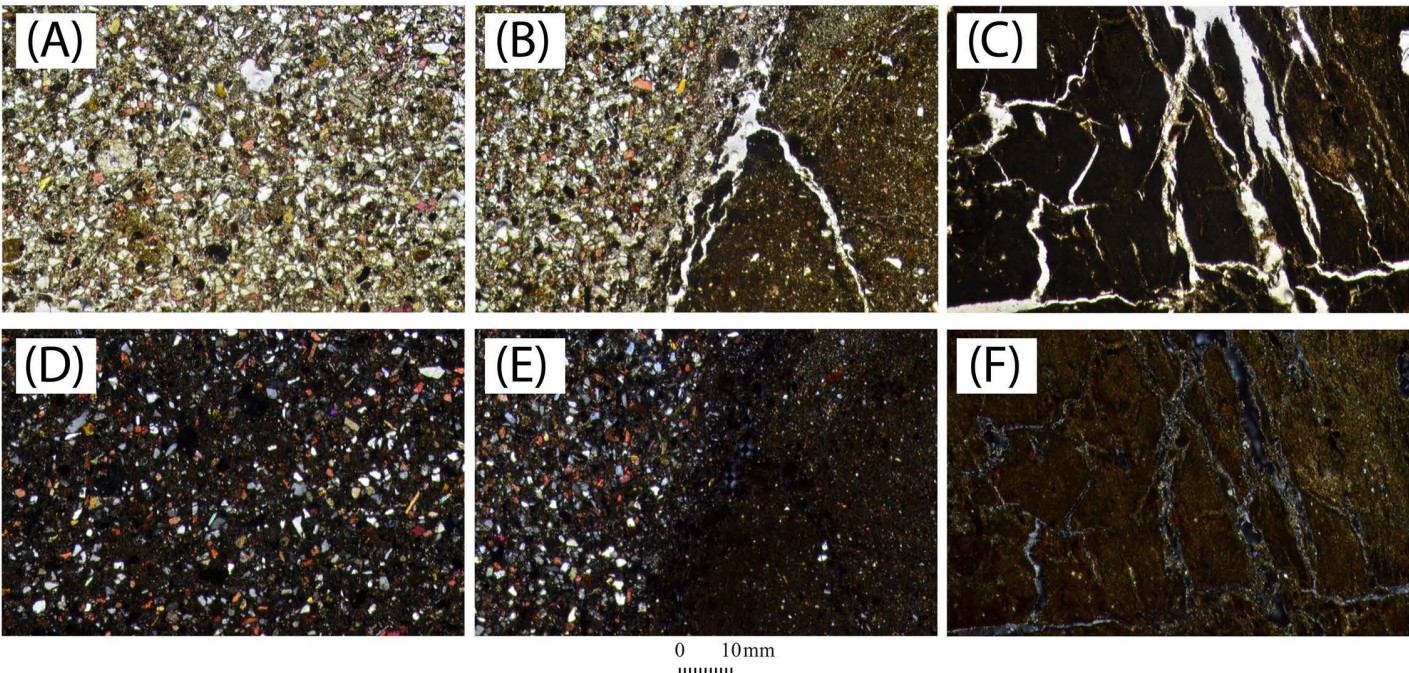

**Fig 9. Selection of the polarised optical microscopy (POM) images.** Three representative samples (nos. 62, 54, 68, respectively) under plane-polarized light (A, B, C) and cross-polarized light (D, E, F).

HL (sample no. 79) and 300.9 HL (sample no. 51). Even after excluding 2 higher and 2 lower values, samples nos. 79 and 51 remain, respectively, the weakest (179.3 HL) and the strongest (286.3 HL). Samples nos. 43, 52, 55, and 67 are also very weak in terms of surface hardness. Without the two higher values, these samples present all values < 200 HL. Based on the mean of the six higher values (threshold 230 HL), samples nos. 45, 46, 49, 57, 59, 60, 63, 68, and 80 also present a strong surface hardness.

Correlating surface cohesiveness with surface hardness test data reveals that samples nos. 49, 59, and 63 gave the best results in both tests, having the highest hardness and the highest surface cohesion, while sample no. 55 is one with the lowest surface cohesion values and the lowest hardness.

## Hydric and rain resistance tests

No static contact angle was identified for any of the 30 samples on either fresh section or actual surfaces. The first contact angle was measured for only a few samples on actual surfaces (samples nos. 43, 44, 46, 49, 53, 58, and 63). Regarding the imbibition time test, we generally observe that the imbibition times (s) are higher on actual surfaces than on fresh sections. Time values of both surfaces and fresh sections are generally below 1 s, indicating the samples are highly porous. In only two cases (samples nos. 44, and 46) is the imbibition time higher than 6 s (S4 Table).

When the hydric test was performed on cubes of approximately 2 cm per side used for the rain resistance test (samples nos. 50, 52, 55, 62, and 79), which were previously saw-cut and sanded, similar results were obtained. Possibly due to sanding, the imbibition time is slightly higher than the first test (> 1 s) for all the samples, except for samples nos. 62_1, 62_2, and 79_2. The cube that shows the highest imbibition values is sample no. 52, with a time comprised between 2.8 and 6.3 s. The sample that shows the lowest imbibition values is sample no. 62, with a time comprised between 0.5 and 0.8 s.

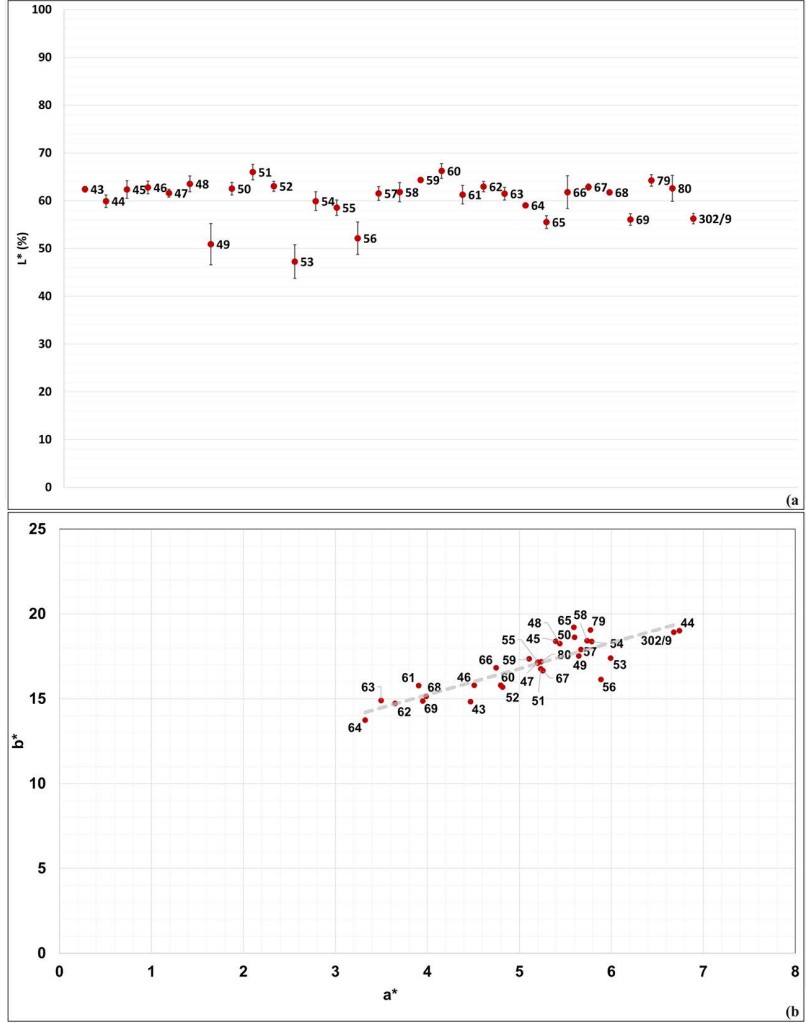

**Fig 10. Chromatic parameters of the 30 analysed samples.** A) L* (mean); B) a*-b* (mean).

The rain resistance test, conducted over four days and simulating one year of rainfall variations, highlighted the critical weakness of adobe bricks when exposed to decay agents, specifically rainfall. Test results (Fig 12) show progressive material disaggregation coinciding with water application (steps 0, 2, and 3). Disaggregation leads to mass loss (Δg), which is directly proportional to the quantity (mL) of applied water, according to the different seasons reproduced in the laboratory.

In terms of cycle duration, sample no. 79 shows clear signs of disaggregation and mass loss from the first water application, and already after the first cycle, it loses almost half of its size and volume (ΔV), while it is the only sample that shows a significant expansion in density (Δρ) due to water absorption. After cycle 2, the sample has already lost all its material cohesion, resulting in a formless mass of clay. Samples nos. 50, 52, and 55 show better results in terms of cohesiveness against weathering agents, with a progressive degradation after each water cycle. They show consistent ΔM, ΔV, and Δρ variations until step 3, while the winter water application has resulted in a great reduction of ΔM and ΔV, accompanied by a significant increase in Δρ for sample no. 52. At the end of the test, sample nos. 50, 52, and 55 have lost all their cohesiveness. The brick with the best water resistance is sample no. 62, which exhibits the most consistent Δ values,

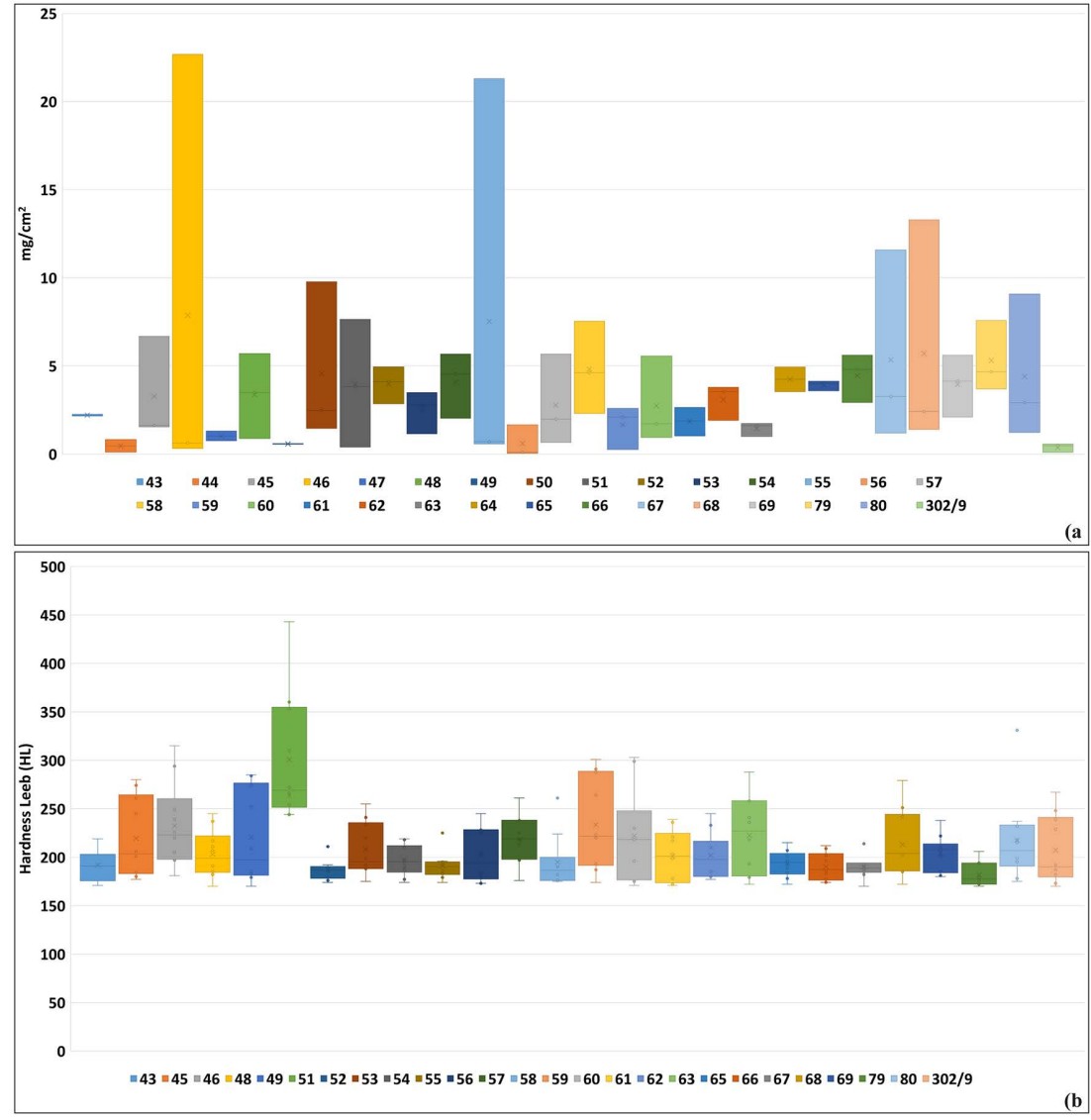

**Fig 11. Surface compactness and strength graphs.** A) Scotch Tape test plot; B) Surface Hardness test plot.

retaining its cubic shape while showing a slow but progressive loss of material with each water cycle. Moreover, during the water application, sample no. 62 is the only one to have shown minimal clay material loss.

Results were standardised by calculating volume and mass variations as if the cubes had sides of exactly 2 cm. This was done to parametrise the data and to allow for comparisons (Fig 13).

## Discussion

Scientific analyses, combined with archaeological data, yielded significant results regarding sample composition and durability in decay tests.

From an archaeological perspective, the autoptic distinction into groups also shows a high degree of homogenisation, with a majority group of 15 out of 30 samples (Gr. 1), most of which are included within the largest group of samples

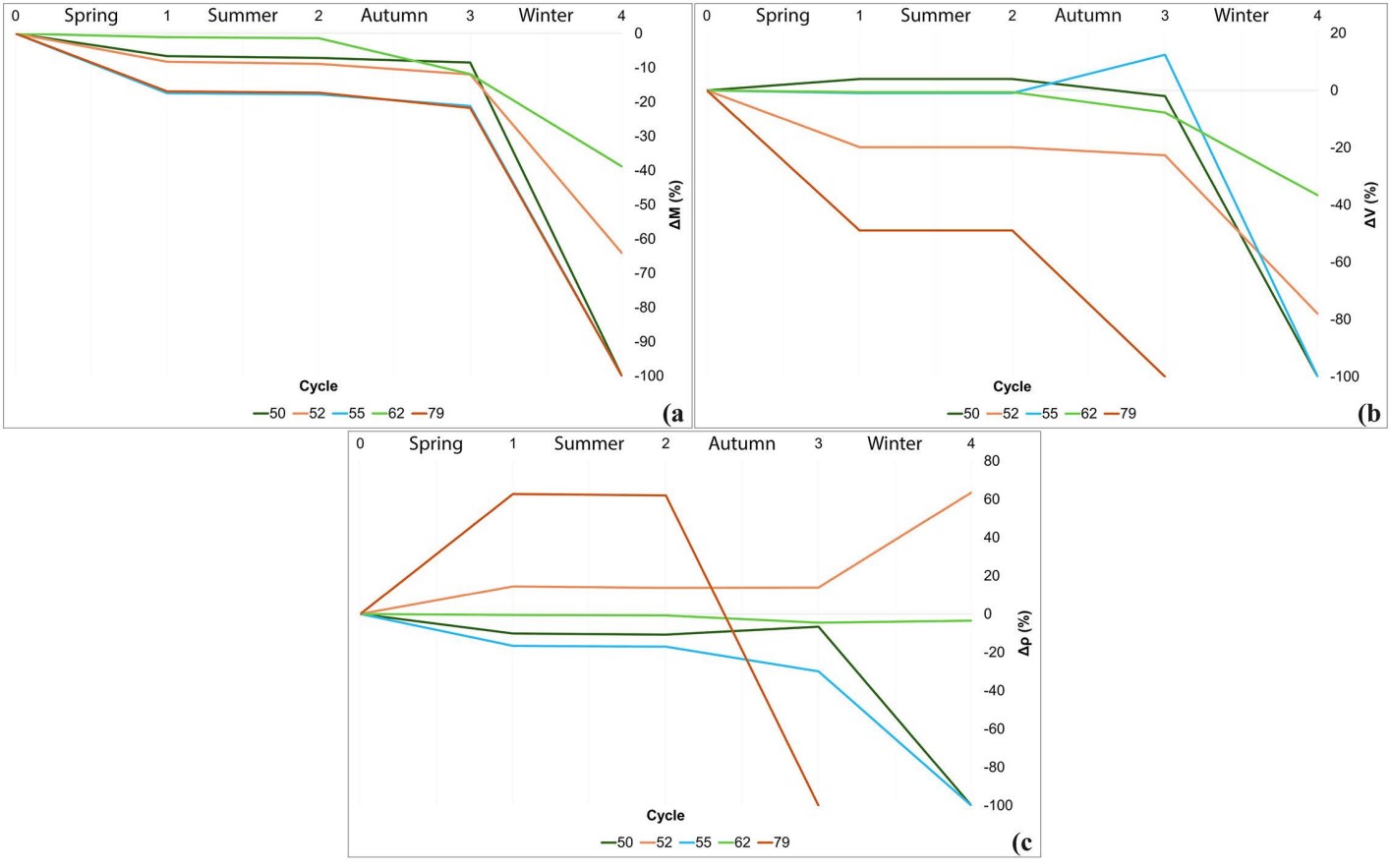

**Fig 12. Rain resistance test.**ΔM (mass), ΔV (volume) and Δρ (density) modification through cycles.

identified by the chemical PCA analysis. The mineralogical data confirms this pattern, with the samples exhibiting strong compositional homogeneity featuring mineral phases consistent with the local geology. Notably, some interesting correlations between archaeological provenience and scientific data can be identified. On a synchronic level, specific associations between the use of particular clays and individual architectural phases identified stratigraphically can be made for the walls of phase AP 2b (Area B, Late Ubaid period; samples nos. 61, 62, and 63) and the 3rd millennium BCE wall W.458 (Area E, ED III/early Akkadian period; samples nos. 79, 80, and earthen plaster sample no. 302/9). The three samples from phase AP 2b all belong to the autoptic Gr. 4 and show a clear correlation in terms of colour measurements – samples all fall between a* (mean) values of 3–4, and b* (mean) values of 13–16, with chemical PCA data showing differentiation of these samples from the majority group based on higher concentration of CaO. On the other hand, samples nos. 79, 80, and 302/9 all belong to Gr. 6 and, mineralogically, show an absence of amphiboles, serpentine, palygorskite, and pyroxenes. No other clear correspondences appear between the autoptic archaeological groups, the colour measurements, and the chemical and mineralogical data.

The homogeneity visible from the mineralogical and chemical data is not confirmed by the petrographic analysis, which reveals diverse preparations based on varying proportions of clay matrix and aggregate content. At least three sub-groups can be distinguished: samples nos. 43, 50, 52, 68, and 79 (including earthen plaster sample no. 302/9) are characterised by a clay-rich matrix with high porosity, attributed to the use of plant fibres as binders and to enhance drying capacity during adobe production; sample no. 54 exhibits a distinct division between a zone containing abundant inert materials

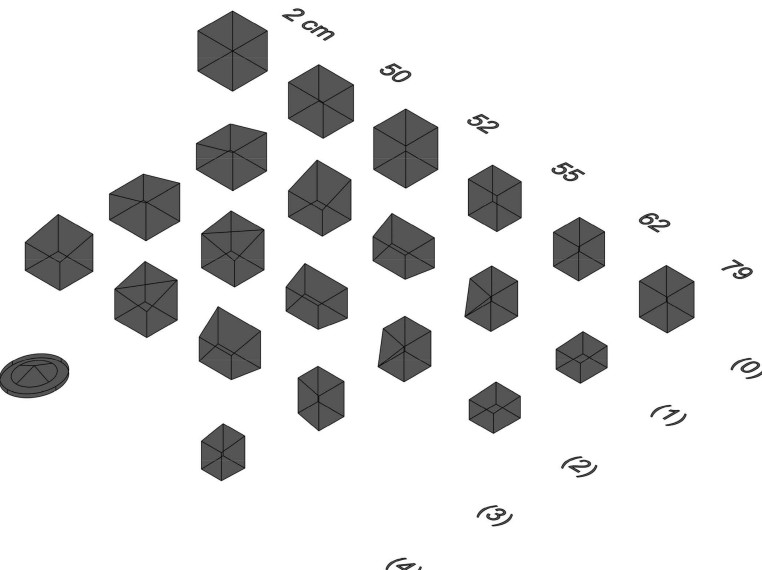

**Fig 13. Rain resistance test.** Result visualisation.

and a zone with a high clay-rich matrix; samples nos. 62 and 64 present an inverse situation with a composition rich in mineral inert inclusions and low in clay binder.

Differences in the petrographic features of samples correspond to different responses obtained in the hydric and mechanical tests. Mechanical tests have shown that most of the samples are relatively weak in terms of both surface strength and cohesiveness, with mean hardness values ranging from 181.8 HL (sample no. 79) to 300.9 HL (sample no. 51), and with the majority of the samples (20 out of 30) presenting a material loss comprised between 2–6 mg/cm$^2$ in terms of surface cohesion.

Samples used for the rain resistance test were selected from those with intermediate values in terms of surface hardness and cohesion, except for sample no. 79, which is the weakest in terms of surface hardness. The aim was to assess the effects of water on those adobe bricks that, in terms of surface hardness and cohesion, appeared to be the weakest, and subsequently to trial potential practical solutions which, if effective on these, might also prove successful on adobe bricks that are intrinsically more resistant.

Notably, sample no. 79 has resulted in being not only the weakest in terms of surface hardness, but also the least resistant to water; however, the reason does not necessarily lie in its surface hardness, but rather in its clay-rich composition. Indeed, the rain resistance test has also demonstrated that the sample no. 62, which exhibits the best water resistance, is not so good in terms of surface cohesion and strength, but it has the most unbalanced matrix-to-aggregate ratio (with high aggregate content), as opposed to samples with a rich clay matrix that decayed more rapidly.

The results indicate that the weakest samples, composed of a clay-rich matrix (e.g., samples nos. 50, 52, and 79), are most susceptible to weathering. In contrast, adobe bricks with an unbalanced matrix-to-aggregate ratio favouring high aggregate content (sample no. 62) exhibit superior water resistance.

## Conclusions

This research has significantly enhanced our understanding of earthen construction techniques at the site of Tell Zurghul/ Nigin in southern Mesopotamia.

The results highlight the utilisation of local raw materials over a diachronic period spanning the 5th to the 3rd millennium BCE. Adobe brick samples from the 5th and 4th millennium BCE display a greater variety of mineralogical phases compared to those from the 3rd millennium BCE, suggesting differences in clay sourcing or purification processes during these prehistoric periods, which may be further investigated in the future from a chaîne opératoire perspective.

A second objective was to investigate the chemical-mineralogical features and hydro-mechanical properties of these bricks to characterise their resistance and durability against weathering. The tests revealed weaknesses and hydrophilicity in certain samples, which varied despite similarities in their mineralogical and chemical composition. Differences in textural features resulted in distinct cohesion and strength properties. The results indicate that the weakest samples, characterised by a clay-rich matrix, are most susceptible to weathering, whereas adobe bricks with a high aggregate content exhibits enhanced water resistance and more efficient drainage behaviour, reflecting the influence of matrix-to-aggregate ratios on material performance.

## Supporting information

**S1 Fig. Temperature (T), humidity (RH), and rainfall data from reference years (2021, 2011, and 2001).** Supplementary information in relation to climatic data for the Tell Zurghul/Nigin area from the WorldClim database at https://worldclim.org/.
(TIF)

**S2 Fig. Polarised optical microscopy (POM) images of all analysed samples (top and bottom) under plane-polarized light (//) and cross-polarized light (⊥).**
(TIF)

**S1 Table. Experimental water.** Monthly mean calculation of rain (mm) for reference years (2021, 2011, 2001) and experimental rain (mL) to be applied to the rain resistance test.
(XLSX)

**S2 Table. Spectroscopy values (L\*, a\*, and b\*) for all analysed samples.**
(XLSX)

**S3 Table. Surface Hardness test values for all analysed samples.**
(XLSX)

**S4 Table. Imbibition time (s) were performed through video analysis, and the first contact angle (CA) was measured by the sessile drop method.**
(XLSX)

## Acknowledgments

We are grateful to the Iraqi SBAH (State Board of Antiquities and Heritage) of Baghdad for permission to conduct scientific analysis on the archaeological materials.

## Author contributions

**Conceptualization:** Luca Volpi, Anna Arizzi.

**Data curation:** Luca Volpi.

**Formal analysis:** Luca Volpi, Francesco Santoro De Vico.

**Funding acquisition:** Luca Volpi, Anna Arizzi.

**Investigation:** Luca Volpi, Francesco Santoro De Vico, Nicola Lanzaro.

**Methodology:** Luca Volpi, Francesco Santoro De Vico, Anna Arizzi, Nicola Lanzaro.

**Resources:** Luca Volpi, Anna Arizzi, Davide Nadali.

**Software:** Francesco Santoro De Vico, Anna Arizzi, Nicola Lanzaro.

**Supervision:** Anna Arizzi.

**Validation:** Anna Arizzi, Davide Nadali.

**Visualization:** Luca Volpi, Francesco Santoro De Vico.

**Writing – original draft:** Luca Volpi, Francesco Santoro De Vico.

**Writing – review & editing:** Luca Volpi, Francesco Santoro De Vico, Anna Arizzi, Nicola Lanzaro, Davide Nadali.

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
