## [Decision Letter · Decision Letter 0]

3 Dec 2025

Dear Dr. Volpi,

Thank you for submitting your manuscript to PLOS ONE. After careful consideration, we feel that it has merit but does not fully meet PLOS ONE’s publication criteria as it currently stands. Therefore, we invite you to submit a revised version of the manuscript that addresses the points raised during the review process.

We look forward to receiving your revised manuscript.

Kind regards,

Santanu Banerjee

Academic Editor

PLOS ONE

**Journal Requirements:**

1. When submitting your revision, we need you to address these additional requirements. Please ensure that your manuscript meets PLOS ONE's style requirements, including those for file naming. The PLOS ONE style templates can be found at https://journals.plos.org/plosone/s/file?id=wjVg/PLOSOne_formatting_sample_main_body.pdf and https://journals.plos.org/plosone/s/file?id=ba62/PLOSOne_formatting_sample_title_authors_affiliations.pdf 2. Please include a complete copy of PLOS’ questionnaire on inclusivity in global research in your revised manuscript. Our policy for research in this area aims to improve transparency in the reporting of research performed outside of researchers’ own country or community. The policy applies to researchers who have travelled to a different country to conduct research, research with Indigenous populations or their lands, and research on cultural artefacts. The questionnaire can also be requested at the journal’s discretion for any other submissions, even if these conditions are not met.  Please find more information on the policy and a link to download a blank copy of the questionnaire here: https://journals.plos.org/plosone/s/best-practices-in-research-reporting. Please upload a completed version of your questionnaire as Supporting Information when you resubmit your manuscript. 3. We note that the grant information you provided in the ‘Funding Information’ and ‘Financial Disclosure’ sections do not match.  When you resubmit, please ensure that you provide the correct grant numbers for the awards you received for your study in the ‘Funding Information’ section. 4. Thank you for stating the following financial disclosure: The CIVIS3i-MSCA research project EnEAp – Endangered Earthen Architecture project. New Challenges in the Conservation and Restoration Practices of Archaeological Earthen Masonries in Western Asia (2022-EnEAp-231; PI: Luca Volpi) has received funding from the European Union’s Horizon 2020 research and innovation programme under grant agreement N°101034324. This research was also funded by the MICIU/AEI/10.13039/501100011033 and FEDER, UE, under the project PID2023-146405OB-100 (2024-2027).  Please state what role the funders took in the study.  If the funders had no role, please state: "The funders had no role in study design, data collection and analysis, decision to publish, or preparation of the manuscript." If this statement is not correct you must amend it as needed. Please include this amended Role of Funder statement in your cover letter; we will change the online submission form on your behalf. 5. Thank you for uploading your study's underlying data set. Unfortunately, the repository you have noted in your Data Availability statement does not qualify as an acceptable data repository according to PLOS's standards. At this time, please upload the minimal data set necessary to replicate your study's findings to a stable, public repository (such as figshare or Dryad) and provide us with the relevant URLs, DOIs, or accession numbers that may be used to access these data. For a list of recommended repositories and additional information on PLOS standards for data deposition, please see https://journals.plos.org/plosone/s/recommended-repositories. 6. When completing the data availability statement of the submission form, you indicated that you will make your data available on acceptance. We strongly recommend all authors decide on a data sharing plan before acceptance, as the process can be lengthy and hold up publication timelines. Please note that, though access restrictions are acceptable now, your entire data will need to be made freely accessible if your manuscript is accepted for publication. This policy applies to all data except where public deposition would breach compliance with the protocol approved by your research ethics board. If you are unable to adhere to our open data policy, please kindly revise your statement to explain your reasoning and we will seek the editor's input on an exemption. Please be assured that, once you have provided your new statement, the assessment of your exemption will not hold up the peer review process. 7. We note that Figures 1 and 2 in your submission contain map images which may be copyrighted. All PLOS content is published under the Creative Commons Attribution License (CC BY 4.0), which means that the manuscript, images, and Supporting Information files will be freely available online, and any third party is permitted to access, download, copy, distribute, and use these materials in any way, even commercially, with proper attribution. For these reasons, we cannot publish previously copyrighted maps or satellite images created using proprietary data, such as Google software (Google Maps, Street View, and Earth). For more information, see our copyright guidelines: http://journals.plos.org/plosone/s/licenses-and-copyright. We require you to either present written permission from the copyright holder to publish these figures specifically under the CC BY 4.0 license, or remove the figures from your submission: a. You may seek permission from the original copyright holder of Figures 1 and 2 to publish the content specifically under the CC BY 4.0 license.   We recommend that you contact the original copyright holder with the Content Permission Form (http://journals.plos.org/plosone/s/file?id=7c09/content-permission-form.pdf) and the following text:“I request permission for the open-access journal PLOS ONE to publish XXX under the Creative Commons Attribution License (CCAL) CC BY 4.0 (http://creativecommons.org/licenses/by/4.0/). Please be aware that this license allows unrestricted use and distribution, even commercially, by third parties. Please reply and provide explicit written permission to publish XXX under a CC BY license and complete the attached form.” Please upload the completed Content Permission Form or other proof of granted permissions as an "Other" file with your submission. In the figure caption of the copyrighted figure, please include the following text: “Reprinted from [ref] under a CC BY license, with permission from [name of publisher], original copyright [original copyright year].” b. If you are unable to obtain permission from the original copyright holder to publish these figures under the CC BY 4.0 license or if the copyright holder’s requirements are incompatible with the CC BY 4.0 license, please either i) remove the figure or ii) supply a replacement figure that complies with the CC BY 4.0 license. Please check copyright information on all replacement figures and update the figure caption with source information. If applicable, please specify in the figure caption text when a figure is similar but not identical to the original image and is therefore for illustrative purposes only.The following resources for replacing copyrighted map figures may be helpful: USGS National Map Viewer (public domain): http://viewer.nationalmap.gov/viewer/The Gateway to Astronaut Photography of Earth (public domain): http://eol.jsc.nasa.gov/sseop/clickmap/Maps at the CIA (public domain): https://www.cia.gov/library/publications/the-world-factbook/index.html and https://www.cia.gov/library/publications/cia-maps-publications/index.htmlNASA Earth Observatory (public domain): http://earthobservatory.nasa.gov/Landsat: http://landsat.visibleearth.nasa.gov/USGS EROS (Earth Resources Observatory and Science (EROS) Center) (public domain): http://eros.usgs.gov/#Natural Earth (public domain): http://www.naturalearthdata.com/ 8. We note that Figure 5 in your submission contain copyrighted images. All PLOS content is published under the Creative Commons Attribution License (CC BY 4.0), which means that the manuscript, images, and Supporting Information files will be freely available online, and any third party is permitted to access, download, copy, distribute, and use these materials in any way, even commercially, with proper attribution. For more information, see our copyright guidelines: http://journals.plos.org/plosone/s/licenses-and-copyright. We require you to either present written permission from the copyright holder to publish these figures specifically under the CC BY 4.0 license, or remove the figures from your submission: a. You may seek permission from the original copyright holder of Figure 5 to publish the content specifically under the CC BY 4.0 license.  We recommend that you contact the original copyright holder with the Content Permission Form (http://journals.plos.org/plosone/s/file?id=7c09/content-permission-form.pdf) and the following text:“I request permission for the open-access journal PLOS ONE to publish XXX under the Creative Commons Attribution License (CCAL) CC BY 4.0 (http://creativecommons.org/licenses/by/4.0/). Please be aware that this license allows unrestricted use and distribution, even commercially, by third parties. Please reply and provide explicit written permission to publish XXX under a CC BY license and complete the attached form.” Please upload the completed Content Permission Form or other proof of granted permissions as an "Other" file with your submission.  In the figure caption of the copyrighted figure, please include the following text: “Reprinted from [ref] under a CC BY license, with permission from [name of publisher], original copyright [original copyright year].” b. If you are unable to obtain permission from the original copyright holder to publish these figures under the CC BY 4.0 license or if the copyright holder’s requirements are incompatible with the CC BY 4.0 license, please either i) remove the figure or ii) supply a replacement figure that complies with the CC BY 4.0 license. Please check copyright information on all replacement figures and update the figure caption with source information. If applicable, please specify in the figure caption text when a figure is similar but not identical to the original image and is therefore for illustrative purposes only. 9. If the reviewer comments include a recommendation to cite specific previously published works, please review and evaluate these publications to determine whether they are relevant and should be cited. There is no requirement to cite these works unless the editor has indicated otherwise. 

**Additional Editor Comments:**

The paper requires minor revision for acceptance, as indicated below.

Reviewer #1: This is a well executed material study of Tell Zurghul/Nigin. The work is adequately rationalised and well written. The weakness of the submitted text is in the conservation context which is only superficially mentioned and neve developed in the text. I would suggest reframing the article as a technical study of the site material which is what they have produced. If there is interest in developing the conservation aspects, I have made suggestions throughout the text. Overall though, thank you for diligently sharing your work.

Summary of Review for Manuscript PONE-D-25-51681R1

Reviewer #2:

This study examines the chemical, mineralogical, hydric and mechanical properties of archaeological earthen mudbricks and earthen plasters from the site of Tell Zurghul/Nigin in southern Iraq, in order to explore construction technique and to identify optimal conservation methods.

The archaeological and geological context is concise but precise. The experimental approach and setup, which includes Polarized Optical Microscopy, X-ray Fluorescence, Powder X-ray Diffraction, colorimetry, together with mechanical and hydric testing is described with precision and the experimental results are convincing, as well as their interpretation.

The most notable results, both in terms of originality and in terms of understanding of these kind of materials, are related to the correlation between the various raw materials used for the mudbricks fabrication and the conservation strategies to be used for earthen structures of provenance. Moreover, the study of the correlation between arcaheological, geological and climatic context are of great importance in the study of the material human evolution in prehistorical ages.

The conclusions are consistent with the evidence and arguments presented

and address the main question posed by the authors, the references are appropriate and both tables and figures are of great utility for a full comprehension of the article.

Reviewers' comments:

Reviewer's Responses to Questions

**Comments to the Author**

1. Is the manuscript technically sound, and do the data support the conclusions?

Reviewer #1: Yes

Reviewer #2: Yes

2. Has the statistical analysis been performed appropriately and rigorously?

Reviewer #1: Yes

Reviewer #2: Yes

3. Have the authors made all data underlying the findings in their manuscript fully available?

Reviewer #1: Yes

Reviewer #2: Yes

4. Is the manuscript presented in an intelligible fashion and written in standard English?

Reviewer #1: Yes

Reviewer #2: Yes

**Reviewer #1:** This is a well executed material study of Tell Zurghul/Nigin. The work is adequately rationalised and well written. The weakness of the submitted text is in the conservation context which is only superficially mentioned and neve developed in the text. I would suggest reframing the article as a technical study of the site material which is what they have produced. If there is interest in developing the conservation aspects, I have made suggestions throughout the text. Overall though, thank you for diligently sharing your work.

**Reviewer #2:**  This study examines the chemical, mineralogical, hydric and mechanical properties of archaeological earthen mudbricks and earthen plasters from the site of Tell Zurghul/Nigin in southern Iraq, in order to explore construction technique and to identify optimal conservation methods.

The archaeological and geological context is concise but precise. The experimental approach and setup, which includes Polarized Optical Microscopy, X-ray Fluorescence, Powder X-ray Diffraction, colorimetry, together with mechanical and hydric testing is described with precision and the experimental results are convincing, as well as their interpretation.

The most notable results, both in terms of originality and in terms of understanding of these kind of materials, are related to the correlation between the various raw materials used for the mudbricks fabrication and the conservation strategies to be used for earthen structures of provenance. Moreover, the study of the correlation between arcaheological, geological and climatic context are of great importance in the study of the material human evolution in prehistorical ages.

The conclusions are consistent with the evidence and arguments presented

and address the main question posed by the authors, the references are appropriate and both tables and figures are of great utility for a full comprehension of the article.

**Do you want your identity to be public for this peer review?** For information about this choice, including consent withdrawal, please see our Privacy Policy

Reviewer #1: **Yes:** Ashley M. Lingle

Reviewer #2: **Yes:** Franco Zanini

---

## [Author Response · Author response to Decision Letter 1]

14 Jan 2026

Below, we summarise how all editorial and technical requirements have been addressed.

1. The revised manuscript now fully complies with PLOS ONE’s style requirements.

2. The questionnaire on inclusivity in global research has been uploaded as Supporting Information.

3. The information reported in the Funding Information and Financial Disclosure sections is now fully consistent.

4. The following statement has been added to the manuscript: “The funders had no role in study design, data collection and analysis, decision to publish, or preparation of the manuscript.”

5. The minimal dataset necessary to replicate the study has been uploaded to Figshare (10.6084/m9.figshare.30933299).

6. Data availability has been clearly stated, and the data are now publicly accessible.

7. The copyright for Figures 1, 4, 5, and 6 belongs to the Missione Archeologica Italiana a Nigin, directed by Prof. Davide Nadali (Sapienza Università di Roma) and field-directed by myself, Luca Volpi (Universidad Autónoma de Madrid). As both copyright holders and authors of the present article, we agree to publish these figures under a CC BY 4.0 licence.

For Figure 1, the image is reprinted from an Open Access journal published under CC BY 4.0 license. For the other images, the requested files have been attached. We kindly ask you to confirm whether the figure captions are appropriate.

8. The copyright for Figure 2 belongs to the Iraqi Bulletin of Geology and Mining, issued by the Iraq Geological Survey (GEOSURV, Iraq), Ministry of Industry and Minerals, Baghdad. The journal operates under a Creative Commons Attribution 4.0 International Licence, which permits redistribution and adaptation provided appropriate credit is given. The remixed map used in our article is available at: https://ibgm-iq.org/ibgm/index.php/ibgm/article/view/263.

The licensing statement is reported at the bottom of the webpage. On this basis, we understand that Figure 2 may be published under a CC BY 4.0 licence. We would appreciate confirmation that the corresponding figure caption is acceptable.

9. All reviewer comments have been carefully considered and addressed. We sincerely thank the reviewers for their constructive feedback and interest in our work. In particular, we have fully taken on board Reviewer #1’s comments regarding the conservation aspects of the manuscript. As a result, we have decided to retain the paper strictly as a material study of Tell Zurghul/Nigin. References to conservation work have therefore been removed from the title – which could have been misleading – and systematically eliminated throughout the text. A single reference has been retained solely to clarify that the material study was developed within a broader research framework concerned with adobe conservation and is intended as a baseline, reference analysis.

Given this shift in focus, we did not include the additional references suggested by the reviewer, as they were specifically aimed at expanding the conservation discussion, which has been intentionally excluded from this article.

In line with these revisions, we have slightly modified the title of the paper. Furthermore, we have replaced all occurrences of the term “mudbrick” with “adobe”, for reasons of terminological accuracy, ethical clarity, and inclusivity.

10. The reference list has been thoroughly reviewed and is now complete and accurate.

---

## [Editor Report · Decision Letter 1]

15 Jan 2026

Chemical-mineralogical features and physical properties of archaeological adobe. The evidence from Tell Zurghul/Nigin (Dhi Qar, Iraq)

PONE-D-25-34039R1

Dear Dr. Volpi,

We’re pleased to inform you that your manuscript has been judged scientifically suitable for publication and will be formally accepted for publication once it meets all outstanding technical requirements.

Kind regards,

Santanu Banerjee

Academic Editor

PLOS One

Additional Editor Comments (optional):

Revisions appear fine. The may paper may be accpted.
---

## [Editor Report · Acceptance letter]

PONE-D-25-34039R1

PLOS One

Dear Dr. Volpi,

I'm pleased to inform you that your manuscript has been deemed suitable for publication in PLOS One. Congratulations! Your manuscript is now being handed over to our production team.

Kind regards,

on behalf of

Dr. Santanu Banerjee

Academic Editor

PLOS One